# A Unified Approach to Submodular Maximization Under Noise

**Kshipra Bhawalkar**
Google
Mountain View, CA
kshipra@google.com

**Yang Cai**
Yale University
New Haven, CT
yang.cai@yale.edu

**Zhe Feng**
Google
Mountain View, CA
zhef@google.com

**Christopher Liaw**
Google
Mountain View, CA
cvliaw@google.com

**Tao Lin**[*]
Harvard University
Cambridge, MA
tlin@g.harvard.edu

## Abstract

We consider the problem of maximizing a submodular function with access to a *noisy* value oracle for the function instead of an exact value oracle. Similar to prior work [13, 16], we assume that the noisy oracle is persistent in that multiple calls to the oracle for a specific set always return the same value. In this model, Hassidim and Singer [13] design a $(1 - 1/e)$-approximation algorithm for monotone submodular maximization subject to a cardinality constraint and Huang et al. [16] design a $(1 - 1/e)/2$-approximation algorithm for monotone submodular maximization subject to any arbitrary matroid constraint. In this paper, we design a meta-algorithm that allows us to take any "robust" algorithm for exact submodular maximization as a black box and transform it into an algorithm for the noisy setting while retaining the approximation guarantee. By using the meta-algorithm with the measured continuous greedy algorithm, we obtain a $(1 - 1/e)$-approximation (resp. $1/e$-approximation) for monotone (resp. non-monotone) submodular maximization subject to a matroid constraint under noise. Furthermore, by using the meta-algorithm with the double greedy algorithm, we obtain a $1/2$-approximation for unconstrained (non-monotone) submodular maximization under noise.

## 1 Introduction

Submodular maximization is a fundamental problem that frequently appears in various forms in many fields such as machine learning, combinatorial optimization, and economics. Submodular functions are functions that satisfy the *diminishing returns property*. More formally, a function $f: 2^N \to \mathbb{R}$ on a ground set $N$ is said to be submodular if for any sets $S \subseteq T \subseteq N$ and element $i \in N \setminus T$, we have $f(S \cup \{i\}) - f(S) \geq f(T \cup \{i\}) - f(T)$. Despite the intuitive nature of submodular functions, many basic problems are NP-hard such as maximizing a monotone submodular function subject to a cardinality constraint or maximizing a non-monotone submodular function without any constraints. Given the inherent intractability of this problem, there has been a vast body of research that aims to develop computationally efficient approximation algorithms for maximizing submodular functions in various settings.

The standard model in submodular maximization assumes that we have *value oracle access* to the submodular function $f$ where, given a set $S$, we can retrieve the exact value of $f(S)$. However, in

---

[*]Work done as a student researcher at Google in 2024.

39th Conference on Neural Information Processing Systems (NeurIPS 2025).

many settings it is not realistic to assume the existence of an exact value oracle. For example, in machine learning, we can never evaluate the true loss function of a model as we do not have access to the true distribution of the population. At best, we may have access to a noisy version of the loss function. In this paper, we study a model for noisy submodular maximization introduced by Hassidim and Singer [13] where querying a set $S$ returns only an unbiased estimate of $f(S)$. A notable feature of this model is *persistent noise*, where querying the value of a set twice returns the same value. Thus, simply querying a submodular function multiple times at the same input cannot be used to denoise the function. Nonetheless, one can still ask whether existing algorithms work in this setting. However, Hassidim and Singer [13] show that this is probably unlikely. For example, they show that the natural greedy algorithm is too sensitive to noise and only obtains an $o(1)$-approximation despite being the optimal algorithm when there is no noise. Thus, new algorithms are needed to deal with noise.

For the problem of maximizing a noisy monotone submodular function subject to a cardinality constraint, Hassidim and Singer [13] designed an algorithm that achieves a tight $(1 - 1/e)$-approximation. At a high-level, their algorithm computes a smooth surrogate function which essentially averages a few queries together to obtain an estimate that may have less noise but potentially be biased. They then apply the greedy algorithm on this surrogate function to obtain a $(1 - 1/e)$-approximation. In a follow-up work, Huang et al. [16] proved a $(1 - 1/e)/2$-approximation ratio for noisy monotone submodular maximization under an arbitrary matroid. This approximation ratio can be improved to $1 - 1/e$ if the matroid satisfies a "strongly base-orderable" property. While Huang et al. [16] also make use of the smoothing technique, their algorithm is based on a local search algorithm [10].

**Main question.** While prior work has established that submodular maximization is feasible in the noisy setting, a notable downside is that the algorithms and analyses are designed for their specific problems. For example, Hassidim and Singer [13] only obtain results for monotone submodular maximization subject to a cardinality constraint and, while Huang et al. [16] extend to arbitrary matroids, their analyses have a factor of 2 gap with the optimal result, despite an algorithm which is provably optimal in the non-noisy setting. Our main question is whether such specific analyses are necessary. In particular, we want to answer the following question:

> *Can we design a framework for noisy submodular maximization that allows us to reuse existing algorithms for the noiseless submodular maximization problems while retaining existing guarantees?*

**Our contributions.** Our main contribution in this paper is an affirmative answer to the above question. In particular, we show that any algorithm for exact submodular optimization which is sufficiently "robust" can be translated to an algorithm for noisy submodular optimization with only an $o(1)$ loss in the approximation ratio. For many settings, this obviates the need for designing new algorithms and instead relies only on checking whether existing algorithms are sufficiently robust. Our technique builds on but differs from [13]: we propose the use of a random surrogate function, which circumvents the limitation of a deterministic surrogate function in [13].

For our first application, we instantiate our framework with the continuous greedy algorithm [4, 9]. Specifically, we use the measured continuous greedy algorithm of Feldman et al. [9] which gives a $(1 - 1/e)$-approximation for monotone submodular maximization and a $1/e$-approximation for non-monotone submodular maximization, both subject to matroid constraints. As our framework inherits the approximation guarantees of these algorithms, we achieve the same approximation ratios for the noisy setting, with only $o(1)$ loss in the approximation ratio. Our second application is to instantiate our framework with the double greedy algorithm [3] to obtain a $1/2$-approximation for noisy unconstrained submodular maximization, which is tight even without noise. To our knowledge, we provide the first set of results for noisy submodular maximization for non-monotone functions, in both unconstrained and matroid constrained settings. We summarize our results in Table 1.

**Other related works.** Another work that is closely related to ours is Hassidim and Singer [14], which also studies the setting where the noise is persistent. In particular, they considered intersection of $P$ matroids and show a $1/(1 + P)$-approximation. There have also been works that look at the noisy submodular setting but *without* persistent noise: for example, Singla et al. [20] consider a model where one only has preference information, and Chen et al. [5] assume that one has access to noisy marginal estimates instead of a value oracle.

Table 1: Our work compared with previous works on noisy submodular maximization. (tight) means optimal approximation ratios achievable by polynomial-time algorithms even without noise.

| Submodular function | Constraint | Previous work | Our work |
|---|---|---|---|
| Monotone | Cardinality | $1 - 1/e$ (tight) [13] | $1 - 1/e$ (tight) Theorem 4.2 |
| | Matroid | $(1 - 1/e)/2$ [16] | $1 - 1/e$ (tight) Theorem 4.2 |
| Non-monotone | Unconstrained | None | $1/2$ (tight) Theorem 4.5 |
| | Matroid | None | $1/e$ Theorem 4.3 |

Another related line of work is optimizing approximately submodular functions, where the noise is not unbiased and can be adversarial. Horel and Singer [15] show that even there is only $1/\sqrt{n}$ noise, the problem becomes intractable and prove exponential lower bounds on the query complexity. On the other hand, they show that constant approximation algorithms are achievable when the "curvature" is bounded or when the noise is small in terms of the rank of the matroid. Zheng et al. [25] study maximizing approximately $k$-submodular functions. Chierichetti et al. [6] study functions that satisfy submodularity $\varepsilon$-approximately and convert them to exactly submodular functions that are $\varepsilon n^2$-close to the original functions, establishing a reduction to the exact submodular setting. A difference with our work is that our noisy value function is far from being $\varepsilon$-approximately submodular. A key contribution of our work is to convert our noisy value function to a surrogate function that is close to the underlying submodular function, which then enables a reduction similar to [6].

A less related line of work is on *online* submodular maximization [12, 17, 18, 21, 22]. Here, one is given a sequence of submodular functions $f_1, f_2, \ldots$ and *before* seeing $f_t$, a player must decide on a set to play. The goal is to minimize "$\alpha$-regret" which is the difference between the value obtained by the player and an $\alpha$-approximation of the optimum. Although the problem setting is different, these works also tend to show that many known algorithms are "robust" and develop a way to reuse these algorithms in the online setting. Such an approach is similar to ours as we show that a number of algorithms are robust and reuse them for the noisy setting.

## 2  Preliminaries

**Submodular set function.**  Let $N$ be a ground set of size $|N| = n$. For a set $S$ and an element $x$, we denote $S + x = S \cup \{x\}$ and $S - x = S \setminus \{x\}$. Let $f : 2^N \to \mathbb{R}_{\geq 0}$ be a non-negative function defined on subsets of $N$. Assume $f(\emptyset) = 0$.[2] We use $f_S(x) = f(S + x) - f(S)$ to denote the marginal value of element $x \in N$ with respect to set $S \subseteq N$. The function $f$ is said to be:

- *submodular* if for any subsets $A \subseteq B \subseteq N$ and any element $x \in N \setminus B$, $f_A(x) \geq f_B(x)$.
- *monotone* if for any subsets $A \subseteq B \subseteq N$, $f(A) \leq f(B)$.

This work considers both monotone and non-monotone (general) submodular functions.

**Noisy value oracle.**  A noisy value oracle for $f$ is denoted by $\tilde{f} : 2^N \to \mathbb{R}_{\geq 0}$. Following previous works [13, 16], we consider a multiplicative noisy value oracle defined by $\tilde{f}(S) = \xi_S f(S)$ for all sets $S \subseteq N$, where $\xi_S$ is a non-negative random variable distributed according to some distribution $\mathcal{D}$. We call $\xi_S$ the noise multiplier for set $S$. Following [13, 16], we assume that the noisy value oracle satisfies the following three properties:

- *Unbiased*: $\mathbb{E}[\tilde{f}(S)] = f(S)$, namely, $\mathbb{E}[\xi_S] = 1$.

---

[2]For our applications, this is without loss of generality since one can add a "dummy" element $x_0$ and then define $g(\emptyset) = 0$, $g(x_0) = f(\emptyset)$, and $g(S) = g(S + x_0) = f(S)$ for $S \neq \emptyset$. The function $g$ remains submodular, and an $\alpha$-approximation for maximizing $g$ gives an $\alpha$-approximation for maximizing $f$. For feasibility, any set that was feasible for the original problem remains feasible after adding $x_0$. This remains a matroid constraint.

- *Persistent*: querying a set $S$ multiple times returns the same value $\tilde{f}(S)$.

- *Independent across different sets*: for different sets $S_1, \ldots, S_k$, $\tilde{f}(S_1), \ldots, \tilde{f}(S_k)$ are independent. Namely, $\xi_{S_1}, \ldots, \xi_{S_k}$ are independent.

If one does not assume persistence, and querying $\tilde{f}(S)$ multiple times gives independent estimates of $f(S)$, then the problem becomes trivial since one can easily estimate $f(S)$ by repeated sampling.

In this work, we assume that the noise multiplier $\xi_S \sim \mathcal{D}$ is sub-exponential.

**Definition 2.1** (see, e.g., Wainwright [24]). *A distribution $\mathcal{D}$ (or a random variable $\xi \sim \mathcal{D}$) is sub-exponential with parameters $(\nu, \alpha)$ if $\mathbb{E}[e^{\lambda(\xi - \mathbb{E}[\xi])}] \leq e^{\frac{\nu^2 \lambda^2}{2}}$ holds for any $\lambda$ satisfying $|\lambda| \leq \frac{1}{\alpha}$.*

Sub-exponential distributions are a large class of distributions, including bounded, Gaussian, and exponential distributions. For example, a random variable $\xi$ bounded in $[0, B]$ is sub-exponential with parameters $(\nu = B, \alpha = 0)$. Moreover, sub-exponential distributions include the generalized exponential tail distributions that have been considered by previous work on monotone submodular maximization under noise [13, 16]. We assume that the parameters $(\nu, \alpha)$ of the sub-exponential distribution $\mathcal{D}$ are known while the distribution $\mathcal{D}$ itself is unknown.

**Matroid constraints.** We aim to maximize the function $f$ using noisy value oracle $\tilde{f}$ over subsets $S \subseteq N$ that satisfy some constraints. Let $\mathcal{I} \subseteq 2^N$ be a collection of feasible subsets of $N$. We assume $\mathcal{I}$ to be downward-closed: i.e., for $I \in \mathcal{I}$ and $I' \subseteq I$, $I' \in \mathcal{I}$. We define two types of constraints:

- *Unconstrained:* $\mathcal{I} = 2^N$.

- *Matroid constraint:* $\mathcal{I}$ is called a matroid if, in addition to being downward-closed, the following condition holds: for any $I_1, I_2 \in \mathcal{I}$ satisfying $|I_1| < |I_2|$, there exists an element $e \in I_2 \setminus I_1$ such that $I_1 \cup \{e\} \in \mathcal{I}$. Each $I \in \mathcal{I}$ is called an independent set. A maximal independent set is called a basis of the matroid. It can be shown that all maximal independent sets are of the same size, which is called the *rank* of the matroid. We denote the rank by $r = r(\mathcal{I})$.

An important special case of a matroid constraint is a cardinality constraint, where $I \in \mathcal{I}$ if and only if $|I| \leq r$.

Given a feasibility set $\mathcal{I}$, we use $O^*$ to denote an optimal set. In other words, $O^* \in \arg\max_{S \in \mathcal{I}} f(S)$. We will use $f(O^*)$ to denote the optimal value. Recall that solving the above optimization problem is generally NP-hard but efficient approximation algorithms are known for many settings.

## 3 A Unified Approach to Noisy Submodular Maximization

In this section, we present a unified approach to the noisy submodular maximization problem. This approach is a reduction to the submodular maximization problem with the exact value oracle.

**Theorem 3.1** (Informal). *Let $\mathcal{A}$ be a "robust" algorithm that obtains an $\alpha$-approximation ratio to the problem $\max_{S \in \mathcal{I}} f(S)$ with exact value oracle. Then, $\mathcal{A}$ can be converted into an algorithm achieving an $(\alpha - o(1))$-approximation to the problem $\max_{S \in \mathcal{I}} f(S)$ with the noisy oracle $\tilde{f}$.*

The definition of "robust" is in Section 3.2 and the formal theorem is Theorem 3.4. We remark that the conversion process above is agnostic to the algorithm $\mathcal{A}$ and just uses $\mathcal{A}$ as a black box.

Our reduction uses an idea proposed by previous work on noisy submodular maximization [13, 16]. This idea is to estimate the value $f(S)$ of a set $S$ using the noisy value of some *surrogate function* $F(S) = \frac{1}{|\mathcal{T}_S|} \sum_{T \in \mathcal{T}_S} f(T)$ where $\mathcal{T}_S$ is a collection of sets related to $S$. One example is to choose a small set $H$ and let $\mathcal{T}_S = \{S \cup H'$ for $H' \subseteq H\}$. In previous work, different surrogate functions were constructed for different settings. Here, we construct a single surrogate function for all the settings we consider (unconstrained non-monotone maximization, monotone and non-monotone maximization under matroid constraints); see Section 3.1 for details. Then, the reduction is to run the algorithm $\mathcal{A}$ to maximize the *noisy* surrogate function $\tilde{F}(S) = \frac{1}{|\mathcal{T}_S|} \sum_{T \in \mathcal{T}_S} \tilde{f}(T) \approx F(S)$.

The main technical challenge here is to ensure that the surrogate function $F(S)$ approximates the true function $f(S)$ well. As shown by [13, 16], a deterministic surrogate function does not always guarantee a good approximation to the true function. So, they need to use arguments that are specific

to their algorithms to show that the surrogate functions work well. A key insight in our work is the use of a *random* surrogate function. We show that such a random surrogate function can approximate the true function in all settings with matroid constraints (see Section 3.3).

## 3.1 Surrogate Function

Let $H \subseteq N$ be a subset of size $|H| = h$ where $h$ is a small integer. We call $H$ a *smoothing set*. Let $t < h$ be another integer. Let $H[t] = \{H' \subseteq H : |H'| = t\}$ be all the subsets of $H$ of size $t$; there are $\binom{h}{t}$ such subsets. We use $H' \sim H[t]$ to denote sampling a subset $H' \subseteq H$ of size $t$ uniformly at random. Define a *surrogate function* $F^{H,t}$ as follows:

$$F^{H,t}(S) = \mathbb{E}_{H' \sim H[t]}\big[f(S \cup H')\big] = \frac{1}{\binom{h}{t}} \sum_{H' \in H[t]} f(S \cup H'), \qquad \forall S \subseteq N. \tag{1}$$

The surrogate marginal value of an element $x$ with respect to $S$ is $F_S^{H,t}(x) = F^{H,t}(S+x) - F^{H,t}(S)$.

**Remark 3.1.** *Another natural way to define a surrogate function would be to take the expectation over all subsets of $H$ and not just subsets of size $t$. This also works when the submodular function $f$ is monotone. Intuitively, this is because adding elements could never hurt. However, when $f$ is non-monotone, adding elements* may *degrade the value of $f$. We thus want to ensure that $\binom{h}{t}$ is large enough for denoising but $t$ is small enough to limit the potential degradation.*

**Claim 3.2.** *The surrogate function $F^{H,t}$ is submodular.*

Because we do not have access to $f$, we cannot query the surrogate function $F^{H,t}$ directly. Instead, we can query the *noisy surrogate function*:

$$\tilde{F}^{H,t}(S) = \frac{1}{\binom{h}{t}} \sum_{H' \in H[t]} \tilde{f}(S \cup H'), \qquad \forall S \subseteq N \tag{2}$$

and the *noisy surrogate marginal value* $\tilde{F}_S^{H,t}(x) = \tilde{F}^{H,t}(S + x) - \tilde{F}^{H,t}(S)$. When $\binom{h}{t}$ is large, $\tilde{F}^{H,t}(S)$ is expensive to compute exactly. Instead, we can approximately compute $\tilde{F}^{H,t}(S)$ by sampling $m$ sets $H_1, \ldots, H_m \sim H[t]$ and taking the sample average:

$$\hat{F}^{H,t,m}(S) = \frac{1}{m} \sum_{i=1}^{m} \tilde{f}(S \cup H_i). \tag{3}$$

To guarantee a good concentration property of $\frac{1}{m} \sum_{i=1}^{m} \tilde{f}(S \cup H_i)$, we sample $H_1, \ldots, H_m \sim H[t]$ *without replacement* to ensure that they are different sets, so $\tilde{f}(S \cup H_1), \ldots, \tilde{f}(S \cup H_m)$ are independent. The following lemma shows that, with high probability, the sample average $\hat{F}^{H,t,m}(S)$ is close to the noisy surrogate value $\tilde{F}^{H,t}(S)$ and the noisy surrogate value $\tilde{F}^{H,t}(S)$ is close to the true surrogate value $F^{H,t}(S)$, when the parameters $h, t, m$ satisfy some condition:

**Lemma 3.3.** *Let $f_{\max} \geq \max_{S \subseteq N} f(S)$ be an upper bound on the maximum value of $f$. Suppose the noise distribution $\mathcal{D}$ is $(\nu, \alpha)$-sub-exponential. Suppose the integers $h, t, m$ satisfy the following:*

$$m \geq \max\{2, 8\nu^2\} \frac{f_{\max}^2}{\varepsilon^2}\big(n + \log \tfrac{4}{\delta}\big), \quad t \geq \log_2(4m), \quad and \quad h = t^2. \tag{4}$$

*Then, for $0 \leq \varepsilon \leq \frac{2\nu^2}{\alpha} f_{\max}$, we have:*

$$\Pr\left[\forall S \subseteq N \setminus H, \; \big|\hat{F}^{H,t,m}(S) - F^{H,t}(S)\big| \leq \varepsilon\right] \geq 1 - \delta. \tag{5}$$

The proof of this lemma uses a Hoeffding inequality for sampling without replacement and a concentration analysis for sub-exponential distribution. It is given in Appendix B.1.

## 3.2 A Meta-Algorithm for Noisy Submodular Maximization

We now present a "meta-algorithm" that converts any algorithm $\mathcal{A}$ for submodular maximization with exact value oracle to an algorithm for noisy value oracle. Given a matroid $\mathcal{I}$ and a set $H \subseteq N$, we consider the contraction of $\mathcal{I}$ by $H$ defined as

$$\mathcal{I}_H = \Big\{S \subseteq N \setminus H \; : \; S \cup H \in \mathcal{I}\Big\} \subseteq \mathcal{I}. \tag{6}$$

In other words, $\mathcal{I}_H$ is the matroid where the independent sets are all sets whose union with $H$ are independent in $\mathcal{I}$. The meta-algorithm (Algorithm 1) works as follows: pick an arbitrary basis $B_0$ of the original matroid $\mathcal{I}$, randomly sample a subset $H$ of $B_0$ of size $h$, run $\mathcal{A}$ to maximize the approximate noisy surrogate function $\hat{F}^{H,t}(S)$ over the minor matroid $\mathcal{I}_H$ to obtain a solution $S_H$, and finally return the set $S_H \cup H'$ where $H'$ is a random subset of $H$ of size $t$.

---

**Algorithm 1:** Meta-algorithm for noisy submodular maximization under matroid constraints

---

**Input**        : Noisy oracle $\tilde{f}$ for a submodular function on ground set $N$. Matroid $\mathcal{I}$.
**Parameter** : $h, t, m$.
1  Let $B_0$ be an arbitrary basis of matroid $\mathcal{I}$, which has size $|B_0| = r$.
2  Sample a subset $H$ of $B_0$ of size $h$ uniformly at random.
3  Run a submodular maximization algorithm $\mathcal{A}$ to solve $\max_{S \in \mathcal{I}_H} F^{H,t}(S)$ using oracle $\hat{F}^{H,t,m}$,
    obtaining a solution $S_H \in \mathcal{I}_H$.
4  Sample a subset $H'$ of $H$ of size $t$ uniformly at random.
5  **Return** $S_H \cup H'$.

---

Before presenting the main result for the meta-algorithm, we define the "robustness" of a submodular maximization algorithm. Let $\mathcal{A}$ be an $\alpha$-approximation algorithm for submodular maximization under constraint $\mathcal{I}$ using the exact value oracle $f$, namely: the solution $S_{\mathcal{A}}$ returned by $\mathcal{A}$ satisfies $\mathbb{E}[f(S_{\mathcal{A}})] \geq \alpha \cdot \max_{S \in \mathcal{I}} f(S)$. We say $\mathcal{A}$ is robust if its performance degrades only a little if the exact value oracle is replaced by an approximate oracle. This is formalized in the following definition.

**Definition 3.1** (Robustness). *An $\varepsilon$-approximate oracle for submodular function $f$ is a function $\hat{f} : 2^N \to \mathbb{R}$ that satisfies $|\hat{f}(S) - f(S)| \leq \varepsilon$ for any queried set $S$. Algorithm $\mathcal{A}$ is $\beta(\varepsilon)$-robust against $\varepsilon$-approximate oracle if, when running $\mathcal{A}$ on $\varepsilon$-approximate oracle $\hat{f}$, the returned solution $\hat{S}_{\mathcal{A}}$ satisfies $\mathbb{E}[f(\hat{S}_{\mathcal{A}})] \geq \alpha \cdot \max_{S \in \mathcal{I}} f(S) - \beta(\varepsilon)$.*

We are now ready to present our main result regarding the meta-algorithm.

**Theorem 3.4.** *Suppose $0 < \varepsilon < \frac{2\nu^2}{\alpha} f_{\max}$ and the parameters $h, t, m$ satisfy (4) with $\delta = 1/n$. If algorithm $\mathcal{A}$ has $\alpha$-approximation ratio and is $\beta(\varepsilon)$-robust against $\varepsilon$-approximate oracle, then the expected value of the solution $\mathrm{ALG}$ returned by Algortihm 1 satisfies:*

- *For non-monotone submodular $f$,  $\mathbb{E}[f(\mathrm{ALG})] \geq \alpha\left(1 - \frac{h}{r-h} - \frac{t}{h-t} - \frac{1}{n}\right)f(O^*) - \beta(\varepsilon)$.*

- *For monotone submodular $f$,  $\mathbb{E}[f(\mathrm{ALG})] \geq \alpha\left(1 - \frac{h}{r-h} - \frac{1}{n}\right)f(O^*) - \beta(\varepsilon)$.*

*Let $Q(\mathcal{A})$ be the query complexity of $\mathcal{A}$. The number of queries to $\tilde{f}$ made by Algorithm 1 is $m \cdot Q(\mathcal{A})$.*

### 3.3   Proof of Theorem 3.4

A key step to prove Theorem 3.4 is to show that, with a randomly sampled smoothing set $H$, maximizing the surrogate function $F^{H,t}$ is roughly equivalent to maximizing the original function $f$. This is formalized by the following lemma, which we call a "smoothing lemma".

**Lemma 3.5** (Smoothing lemma). *Let $f$ be a submodular function.*

- *For any $f$, we have  $\mathbb{E}_{H \sim B_0[h]}\left[\max_{S \in \mathcal{I}_H} F^{H,t}(S)\right] \geq \left(1 - \frac{h}{r-h} - \frac{t}{h-t}\right)f(O^*)$.*

- *If $f$ is monotone then  $\mathbb{E}_{H \sim B_0[h]}\left[\max_{S \in \mathcal{I}_H} F^{H,t}(S)\right] \geq \left(1 - \frac{h}{r-h}\right)f(O^*)$.*

*Proof of Theorem 3.4.*  According to Lemma 3.3, with probability at least $1 - \delta$, the function $\hat{F}^{H,t,m}$ is an $\varepsilon$-approximate oracle for $F^{H,t}$ for all sets $S \in \mathcal{I}_H$. We denote this event by $\mathcal{E}$. Conditioning on $\mathcal{E}$, the expected value of the solution $\mathrm{ALG}$ returned by Algorithm 1 satisfies

$$
\begin{aligned}
\mathbb{E}[f(\mathrm{ALG}) \mid \mathcal{E}] &= \mathbb{E}_H \mathbb{E}_{\text{randomness of } \mathcal{A}} \mathbb{E}_{H' \sim H[t]}\left[f(S_H \cup H') \mid \mathcal{E}\right] \\
&= \mathbb{E}_H \mathbb{E}_{\text{randomness of } \mathcal{A}}\left[F^{H,t}(S_H) \mid \mathcal{E}\right] && \text{by the definition of } F^{H,t} \\
&\geq \alpha \cdot \mathbb{E}_H\left[\max_{S \in \mathcal{I}_H} F^{H,t}(S)\right] - \beta(\varepsilon) && \text{by } \beta(\varepsilon)\text{-robustness of } \mathcal{A}.
\end{aligned}
$$

By the smoothing lemma (Lemma 3.5), for any submodular function $f$, we have

$$\mathbb{E}[f(\text{ALG}) \mid \mathcal{E}] \geq \alpha \cdot \left(1 - \frac{h}{r-h} - \frac{t}{h-t}\right) f(O^*) - \beta(\varepsilon).$$

Since the event $\mathcal{E}$ happens with probability at least $1 - \delta$, we have

$$\mathbb{E}[f(\text{ALG})] \geq (1-\delta)\left(\alpha\left(1 - \frac{h}{r-h} - \frac{t}{h-t}\right)f(O^*) - \beta(\varepsilon)\right) + \delta \cdot 0$$

$$\geq \alpha\left(1 - \frac{h}{r-h} - \frac{t}{h-t} - \delta\right)f(O^*) - \beta(\varepsilon).$$

Letting $\delta = \frac{1}{n}$ proves the theorem for the first case. For the monotone case, we can remove the $\frac{t}{h-t}$ term by Lemma 3.5. $\square$

It remains to prove Lemma 3.5.

*Proof of Lemma 3.5.* We prove this lemma for the non-monotone case here. The proof for the monotone case is simpler and given in Appendix B.2. We will use the following *basis exchange property* for matroids.

**Lemma 3.6** (Donald and Tobey [7]). *For any two bases $B_1, B_2$ of a matroid, for any integer $h \geq 1$, there exists a bijection $\sigma$ from subsets of $B_1$ with size $h$ to subsets of $B_2$ with size $h$ such that, for every subset $H \subseteq B_1$ with size $h$, $B_2 - \sigma(H) + H$ is a basis.*

Recall that $O^* = \arg\max_{O \in \mathcal{I}} f(O)$ is an optimal solution for $f$ over the original matorid $\mathcal{I}$. Since $f$ is non-monotone, $O^*$ is not necessarily a basis of matroid $\mathcal{I}$. Let $B_1 \supseteq O^*$ be any basis of matroid $\mathcal{I}$ that contains $O^*$. We apply Lemma 3.6 to bases $B_0$ and $B_1$ to obtain a bijection $\sigma$ between subsets of $B_0$ with size $h$ and subsets of $B_1$ with size $h$, such that $B_1 - \sigma(H) + H$ is a basis of matroid $\mathcal{I}$, for every subset $H \subseteq B_0$ with size $h$. We note that $\left(B_1 - \sigma(H)\right) \cap H = \emptyset$[3], so $B_1 - \sigma(H)$ belongs to $\mathcal{I}_H$. Because a matroid is downward-closed, we have $O^* - \sigma(H) \subseteq B_1 - \sigma(H) \in \mathcal{I}_H$. Thus, $\max_{S' \in \mathcal{I}_H} F^{H,t}(S') \geq F^{H,t}(O^* - \sigma(H))$. Taking expectation over $H \sim B_0[h]$, we get

$$\mathbb{E}_{H \sim B_0[h]}\left[\max_{S \in \mathcal{I}_H} F^{H,t}(S)\right] \geq \mathbb{E}_{H \sim B_0[h]}\left[F^{H,t}(O^* - \sigma(H))\right]$$

$$= \mathbb{E}_H \mathbb{E}_{H' \sim H[t]}\left[f(O^* - \sigma(H) + H')\right] \tag{7}$$

$$\geq \mathbb{E}_H\left[f(O^* - \sigma(H)) - \frac{|H'|}{|H| - |H'|}\max_{S' \subseteq O^* - \sigma(H) + H} f(S')\right] \qquad \text{by Lemma A.3}$$

$$\geq \mathbb{E}_H\left[f(O^* - \sigma(H))\right] - \frac{t}{h-t}f(O^*).$$

Because $\sigma$ is a bijection between subsets of $B_0$ and subsets of $B_1$ and $H$ is a uniformly random subset of $B_0$ with size $h$, $\sigma(H)$ must be a uniformly random subset of $B_1$ with size $h$. By Lemma A.2

$$\mathbb{E}_H\left[f(O^* - \sigma(H))\right] \geq f(O^*) - \frac{h}{|B_1| - h} \cdot \max_{S' \subseteq O^* \cap B_1} f(S') = f(O^*) - \frac{h}{r-h}f(O^*). \tag{8}$$

This implies

$$\mathbb{E}_H\left[\max_{S \in \mathcal{I}_H} F^{H,t}(S)\right] \geq f(O^*) - \frac{h}{r-h}f(O^*) - \frac{t}{h-t}f(O^*),$$

which proves the lemma for the non-monotone case. $\square$

## 4  Noisy Submodular Maximization Under Specific Settings

In this section, we instantiate our meta-algorithm (Algorithm 1) with existing algorithms for submodular maximization with exact value oracle. By proving that existing algorithms are robust, we obtain results for noisy submodular maximization under various specific settings.

---

[3]Otherwise, the size $|B_1 - \sigma(H) + H| < |B_1 - \sigma(H)| + |H| = r - h + h = r$, contradicting the fact that $B_1 - \sigma(H) + H$ is a basis and should have size $r$.

## 4.1 Matroid Constraints

**Robustness of Measured Continuous Greedy.** We first consider maximizing monotone and non-monotone submodular functions under matroid constratins. We prove that the *measured continuous greedy algorithm* of Feldman et al. [9] is robust. Here, we recall the algorithm (with exact value oracle). First, define the multilinear extension of a submodular function $f$ as

$$F(x) = \sum_{S \subseteq [n]} f(S) \prod_{i \in S} x_i \prod_{i \notin S} (1 - x_i), \quad \forall x \in [0, 1]^n.$$

Then the algorithm works as follows. First, define $x(0) = \mathbf{0}$ (the 0 vector) and let $\delta \in (0, 1)$ be such that $1/\delta$ is an integer. Given a point $x_i(t)$ at time $t$, we first solve

$$y^*(t) \in \arg\max_{\mathcal{P}} \left\{ \sum_{i=1}^n \partial_i F(x(t)) y_i \right\},$$

where the $\arg\max$ is taken over the matroid polytope $\mathcal{P}$. Then we update $x_i(t + \delta) = x_i(t) + \delta(1 - x_i(t)) y_i^*(t)$. Finally, we use pipage rounding [4] (which is an obvlious rounding scheme) to convert the fractional solution $x(1)$ to a discrete set $S \in \mathcal{I}$. The pipage rounding technique guarantees that $\mathbb{E}[f(S)] \geq F(x(1))$. In addition, we have that $F(x(1))/f(\text{OPT}) \geq 1 - 1/e - O(n^3 \delta)$ when $f$ is monotone and $F(x(1))/f(\text{OPT}) \geq 1/e - O(n^3 \delta)$ when $f$ is non-monotone.

The following lemma establishes the robustness of the measured continuous greedy algorithm.

**Lemma 4.1** (Robustness of measured continuous greedy)**.** *For submodular function maximization under matroid constraint, the measured continuous greedy algorithm [9] obtains a $(1 - 1/e)$-approximation when $f$ is monotone and a $(1/e)$-approximation when $f$ is non-monotone. Moreover, the algorithm is $O(n\varepsilon)$-robust.*

The proof can be found in Appendix C. In particular, see Lemma C.6 for the monotone case and Lemma C.8 for the non-monotone case. Note that, technically, Lemma 4.1 has a small discretization error but this can be made arbitrarily small so we omit it in the statement. In addition, we can absorb the discretization error into the error due to noise in our theorem statements below.

**Monotone Submodular Functions with Matroid Constraints.** We apply Theorem 3.4 to the problem of maximizing monotone submodular functions with matroid constraints under a noisy value oracle. Let the algorithm $\mathcal{A}$ in Algorithm 1 be the measured continuous greedy algorithm [9] mentioned above. Fix parameter $\varepsilon \in (0, \frac{2\nu^2}{\alpha})$. Choose integer $m \geq \max\{2, 8\nu^2\} \frac{n^4}{\varepsilon^2} (n + \log(4n)) = \tilde{O}(\frac{n^5}{\varepsilon^2})$, $t \geq \log_2(4m) = \Theta(\log(\frac{n}{\varepsilon}))$, and $h = t^2 = \Theta(\log^2(\frac{n}{\varepsilon}))$. Then, we apply Theorem 3.4 and Lemma 4.1 with parameter $\varepsilon_1 = \frac{\varepsilon f_{\max}}{n^2}$ to obtain:

$$
\begin{aligned}
\mathbb{E}[f(\text{ALG})] &\geq \left(1 - \frac{1}{e}\right)\left(1 - \frac{h}{r - h} - \frac{1}{n}\right) f(O^*) - O(n\varepsilon_1) \\
&= \left(1 - \frac{1}{e}\right)\left(1 - \frac{\Theta(\log^2(\frac{n}{\varepsilon}))}{r - \Theta(\log^2(\frac{n}{\varepsilon}))} - \frac{1}{n}\right) f(O^*) - O\left(\frac{\varepsilon f_{\max}}{n}\right) \\
&\geq \left(1 - \frac{1}{e} - \frac{\Theta(\log^2(\frac{n}{\varepsilon}))}{r - \Theta(\log^2(\frac{n}{\varepsilon}))} - \frac{1}{n} - O(\varepsilon)\right) f(O^*),
\end{aligned}
$$

which immediately leads to the following corollary.

**Theorem 4.2.** *Fix $\varepsilon \in (0, \frac{2\nu^2}{\alpha})$. Suppose $n \geq \frac{1}{\varepsilon}$ and the matroid's rank $r \geq \Omega(\frac{1}{\varepsilon} \log^2(\frac{n}{\varepsilon}))$. By letting the $\mathcal{A}$ in Algorithm 1 be the measured continuous greedy algorithm [9], we obtain a polynomial-time algorithm for maximizing monotone submodular functions under matroid constraints with noisy oracle satisfying $\mathbb{E}[f(\text{ALG})] \geq \left(1 - \frac{1}{e} - O(\varepsilon)\right) f(O^*)$.*

Previous work [16] gave a polynomial-time algorithm with $(1 - 1/e)/2 - O(\varepsilon)$ approximation ratio for noisy monotone submodular maximization under matroid constraints, assuming $n \geq \varepsilon^{-4}$ and $r \geq \varepsilon^{-4/3}$. Our Theorem 4.2 improves [16] by increasing the approximation ratio to $1 - 1/e - O(\varepsilon)$ (which is the tight ratio even with the exact value oracle) as well as relaxing the condition on $n$ and $r$.

In the special case of cardinality constraints, [16]'s algorithm achieves $(1-1/e-O(\varepsilon))$ approximation with $\tilde{O}(r^2 n^3/\varepsilon)$ query complexity (their Theorem 4.6). By letting $\mathcal{A}$ be the greedy algorithm with query complexity $Q(\mathcal{A}) = O(rn)$, our algorithm achieves the same approximation ratio with query complexity $m \cdot Q(\mathcal{A}) = \tilde{O}(n^5/\varepsilon^2) \cdot O(rn) = \tilde{O}(rn^6/\varepsilon^2)$. While our query complexity might be worse than [16], the advantage of our algorithm lies in its generality and better approximation ratio in the more general matroid constraints case.

**Non-Monotone Submodular Functions with Matroid Constraint.** For non-monotone submodular functions, choosing the parameters as above, by Theorem 3.4 and Lemma 4.1 we obtain the following:

$$\mathbb{E}[f(\mathrm{ALG})] \geq \frac{1}{e} \cdot \Big(1 - \frac{h}{r-h} - \frac{t}{h-t} - \frac{1}{n}\Big)f(O^*) - O(n\varepsilon_1)$$

$$\geq \Big(\frac{1}{e} - \frac{\Theta(\log^2(\frac{n}{\varepsilon}))}{r - \Theta(\log^2(\frac{n}{\varepsilon}))} - \frac{1}{\Theta(\log\frac{n}{\varepsilon})} - \frac{1}{n} - O(\varepsilon)\Big)f(O^*).$$

**Theorem 4.3.** *Fix $\varepsilon \in (0, \frac{2\nu^2}{\alpha})$. Suppose $n \geq \frac{1}{\varepsilon}$ and the matroid's rank $r \geq \Omega(\frac{1}{\varepsilon}\log^2(\frac{n}{\varepsilon}))$. By letting the $\mathcal{A}$ in Algorithm 1 be the measured continuous greedy algorithm [9], we obtain a polynomial-time algorithm for maximizing non-monotone submodular functions under matroid constraints with noisy oracle satisfying $\mathbb{E}[f(\mathrm{ALG})] \geq \big(\frac{1}{e} - \frac{1}{\Theta(\log\frac{n}{\varepsilon})} - O(\varepsilon)\big)f(O^*)$.*

The $\frac{1}{e} \approx 0.367$ approximation above is not necessarily tight. Buchbinder and Feldman [2] design a $0.401$-approximation algorithm for non-monotone submodular maximization under matroid constraint with exact value oracle. Their algorithm is quite technical, so we leave as an open question whether their algorithm is robust against approximate value oracle. If their algorithm is robust, then it can be directly converted to an algorithm for noisy non-monotone submodular maximization under matroid constraint achieving $0.401 - o(1)$ approximation, using our Algorithm 1.

**High-Probability Result.** While our main results (Theorems 3.4, 4.2 and 4.3) are stated in terms of the expected value $\mathbb{E}[f(\mathrm{ALG})]$, we can obtain a high-probability result for maximizing monotone submodular functions under noise. The idea is to repeat Algorithm 1 multiple times and output the best solution. The challenge here is to compare two sets $S_1, S_2$ using noisy values, without access to the true values $f(S_1), f(S_2)$. For monotone functions, one can construct another surrogate function $\tilde{f}_0(S)$, by randomly removing an element from $S$, to do the comparison. Monotonicity combined with submodularity ensures $f(S) \geq f_0(S) \geq (1 - \frac{1}{|S|})f(S)$. Such an idea was also utilized by [16]. However, constructing such surrogate functions for non-monotone functions becomes technically challenging. The surrogate function above no longer works since removing an element could actually *improve* the objective. Indeed, an approximate version of monotonicity is not true in general. We leave for future work to obtain high-probability results for maximizing non-monotone functions under noise. See Appendix F for a detailed discussion.

## 4.2 Unconstrained Submodular Maximization

Finally, we consider maximizing any (non-monotone) submodular function without constraints. Here, we instantiate Theorem 3.4 with the double greedy algorithm [3] which is known to give a $1/2$-approximation for unconstrained submodular maximization. At a high-level, the double greedy algorithm works as follows. We first initialize two sets $X_0 = \emptyset$ and $Y_0 = N$. Let $N = \{e_1, \ldots, e_n\}$. For $t = 1, \ldots, n$, we check the marginal value $a_t$ of adding $e_t$ to $X_{t-1}$ and the marginal value $b_t$ of removing $e_t$ from $X_{t-1}$. With probability $a_t/(a_t + b_t)$ (with some clipping operations if necessary), we add $e_t$ to $X_{t-1}$ to get $X_t$ and set $Y_t = Y_{t-1}$. Otherwise, we set $X_t = X_{t-1}$ and remove $e_t$ from $Y_{t-1}$ to get $Y_t$. A formal description of the algorithm can be found in Appendix D. The key fact that we require is the following lemma, with proof given in Appendix D.

**Lemma 4.4** (Robustness of double greedy). *The double greedy algorithm obtains $1/2$-approximation for maximizing a submodular function without constraints. Moreover, the algorithm is $O(n\varepsilon)$-robust.*

Directly applying Lemma 4.4 and the non-monotone case in Theorem 3.4 gives us an algorithm for maximizing unconstrained non-monotone functions with noisy oracle with $\frac{1}{2} - \frac{1}{\Theta(\log\frac{n}{\varepsilon})} - O(\varepsilon)$

approximation ratio. Yet, the unconstrained case allows for a more refined analysis which gives a better approximation ratio that removes the $\frac{1}{\Theta(\log \frac{n}{\varepsilon})}$ term (see Appendix E for the proof):

**Theorem 4.5.** *Fix $\varepsilon \in (0, \frac{2\nu^2}{\alpha})$. Suppose $n \geq \Omega(\frac{1}{\varepsilon} \log^2(\frac{n}{\varepsilon}))$. By letting the $\mathcal{A}$ in Algorithm 1 be the double greedy algorithm [3], we obtain a polynomial-time algorithm for maximizing unconstrained non-monotone submodular functions with noisy oracle satisfying $\mathbb{E}[f(\mathrm{ALG})] \geq \left(\frac{1}{2} - O(\varepsilon)\right)f(O^*)$.*

## 5 Simulation Results

In this section, we present simulation results to compare the performances of our proposed algorithm (Algorithm 1) and some heuristic algorithms for the noisy submodular maximization problem.

We focus on the simple yet underexplored case of unconstrained non-monotone noisy submodular maximization. We consider an example where the submodular function is a weighted additive function with quadratic cost in the subset size: $\forall S \subseteq N, f(S) = \sum_{i \in S} w_i - c|S|^2$, where each element $i$ has weight $w_i \sim \mathrm{Uniform}[0, 20]$, with cost parameter $c = 10/n$, so the ground set $N$ has expected value 0. When sampling $w_i$, we ensure that $f$ is non-negative. The noisy value function is $\tilde{f}(S) = \xi_S f(S)$ where $\xi_S \sim \mathrm{Normal}(\mu = 1, \sigma^2 = 0.1)$. We compare four algorithms against the optimal value benchmark $f(O^*) = \max_{S \subseteq N} f(S)$:

- **Double greedy (DG) with exact value oracle** [3]: this is a worst-case optimal polynomial-time algorithm in the noiseless setting. It uses the exact value oracle and is used only for reference.
- **Double greedy (DG) with noisy value oracle**: the DG algorithm that uses the noisy value oracle directly. It is a natural algorithm to compare to, given the optimality of noiseless DG.
- **Random subset**: pick a subset of size $n/2$ uniformly at random.
- **Our algorithm in Theorem 4.5**: our surrogate-value-based meta algorithm (Algorithm 1) instantiated with DG. After simple tuning, we set the parameters to $h = 20, t = 4$, and vary $m$.

We run 1000 simulations. In each simulation, we first sample $f$ (namely, the weights $w_i$) and the noisy function $\tilde{f}$ (the multipliers $\xi_S$), then run each of the above four algorithms once. Table 2 shows the means $\mathbb{E}\left[\frac{f(\mathrm{ALG})}{f(O^*)}\right]$ and standard deviations of the true values of the obtained sets, as a fraction of the optimal value. We observed that the heuristic algorithm, DG with Noisy Oracle, does not perform well; it is only slightly better than Random Subset. Our algorithm significantly outperforms the heuristic algorithms, with $\approx 15\%$ improvement with $m = 50$ and $\approx 25\%$ with $m = 200$.

Table 2: Comparison between ours and other algorithms, in the unconstrained non-monotone noisy submodular maximization setting.

| Ground Set Size | DG with Exact Oracle | DG with Noisy Oracle | Random Subset | Our Algorithm ($m = 50$) | Our Algorithm ($m = 200$) |
|---|---|---|---|---|---|
| $n = 50$ | 0.944 (± 0.028) | 0.601 (± 0.074) | 0.550 (± 0.079) | 0.674 (± 0.067) | 0.735 (± 0.059) |
| $n = 100$ | 0.944 (± 0.019) | 0.565 (± 0.055) | 0.536 (± 0.057) | 0.657 (± 0.047) | 0.731 (± 0.041) |

## 6 Conclusion and Future Work

In this paper, we developed a framework for noisy submodular maximization which allows us to reuse existing robust algorithms for submodular maximization while obtaining essentially the same approximation guarantee. As applications, we considered submodular maximization subject to a matroid constraint (both monotone and non-monotone) and unconstrained non-monotone submodular maximization. We conclude this paper with a few open questions. The first is to obtain high-probability bounds for non-monotone submodular maximization. The second is to explore whether there is some meta-algorithm that can take *any* existing algorithm and utilize it for the noisy setting. Finally, an interesting open direction is to explore noisy submodular *minimization*.

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

# A Useful Facts

**Lemma A.1.** *Let $f$ be a non-negative submodular function. Let $S, A \subseteq N$ be two sets. Let $x$ be a random element from $A$. Then $\mathbb{E}_{x \sim A}[f(S) - f(S - x)] \leq \frac{1}{|A|} f(S)$.*

*Proof.* Denote $S \cap A = \{x_1, \ldots, x_{|S \cap A|}\}$. We have

$$\mathbb{E}_{x \sim A}\Big[f(S) - f(S \setminus \{x\})\Big]$$

$$= \Pr[x \in S \cap A] \cdot \mathbb{E}_{x \sim S \cap A}\Big[f(S) - f(S \setminus \{x\})\Big]$$

$$= \frac{|S \cap A|}{|A|} \cdot \frac{1}{|S \cap A|} \sum_{i=1}^{|S \cap A|} \Big[f(S) - f(S \setminus \{x_i\})\Big]$$

$$\leq \frac{|S \cap A|}{|A|} \cdot \frac{1}{|S \cap A|} \sum_{i=1}^{|S \cap A|} \Big[f(S \setminus \{x_1, \ldots, x_{i-1}\}) - f(S \setminus \{x_1, \ldots, x_{i-1}, x_i\})\Big]$$

$$= \frac{1}{|A|} \big(f(S) - f(S \setminus A)\big)$$

$$\leq \frac{1}{|A|} f(S),$$

as desired. $\qquad\square$

**Lemma A.2.** *For any two sets $S, A \subseteq N$, for integer $k \geq 1$, sampling a subset $B$ of size $k$ from $A$ uniformly at random, we have:*

$$\mathbb{E}_{B \sim A[k]}\big[f(S \setminus B)\big] \geq f(S) - \frac{k}{|A| - k} \cdot \max_{S' \subseteq S \cap A, \, |S'| \geq |S \cap A| - k} f(S').$$

*Proof.* The proof is by induction. Let $B_{i-1}$ be $i - 1$ random elements from $A$ without duplicates. Let $b_i \in A \setminus B_{i-1}$ and $B_i = B_{i-1} \cup \{b_i\}$. By the induction hypothesis, we have

$$\mathbb{E}_{B_{i-1} \sim A[i-1]}\big[f(S \setminus B_{i-1})\big] \geq f(S) - \frac{i-1}{|A| - i + 1} \cdot \max_{S' \subseteq S \cap A, \, |S'| \geq |S \cap A| - i + 1} f(S').$$

We now condition on $B_{i-1}$. By Lemma A.1, we have

$$\mathbb{E}_{b_i}[f(S \setminus B_i)] - f(S \setminus B_{i-1}) \geq -\frac{1}{|A| - i + 1} f(S \setminus B_{i-1}).$$

Adding the last two inequalities gives $\mathbb{E}[f(S \setminus B_i)] \geq f(S) - \frac{i}{|A|-i} \max_{S' \subseteq S \cap A, |S'| \geq |S \cap A| - i} f(S')$. The lemma follows by induction. $\qquad\square$

**Lemma A.3.** *For any two sets $S, A \subseteq N$, for integer $k \geq 1$, sampling a subset $B$ of size $k$ from $A$ uniformly at random, we have:*

$$\mathbb{E}_{B \sim A[k]}\big[f(S \cup B)\big] \geq f(S) - \frac{k}{|A| - k} \max_{S' : S \subseteq S' \subseteq S \cup A} f(S').$$

*Proof.* Follows from Lemma A.2 by applying it to the submodular function $g(S) = f(N \setminus S)$. $\quad\square$

# B Missing Proofs from Section 3

## B.1 Proof of Lemma 3.3

**Lemma B.1** (Hoeffding's inequality for sub-exponential distributions: see, e.g., page 29 in [24]). *Let $X_1, \ldots, X_m$ be independent random variables where each $X_i$ is $0$-mean and $(\nu_i, \alpha_i)$-sub-exponential. Let $\alpha_* = \max_{i=1}^m \alpha_i$ and $\nu_* = \sqrt{\sum_{i=1}^m \nu_i^2}$. Then,*

$$\Pr\Big[\,|\frac{1}{m}\sum_{i=1}^m X_i| \geq \varepsilon\Big] \leq \begin{cases} 2\exp\big(-\frac{m\varepsilon^2}{2\nu_*^2/m}\big) & \text{for } 0 \leq \varepsilon \leq \frac{\nu_*^2}{m\alpha_*}, \\ 2\exp\big(-\frac{m\varepsilon}{2\alpha_*}\big) & \text{for } \varepsilon > \frac{\nu_*^2}{m\alpha_*}. \end{cases}$$

*Proof of Lemma 3.3.* Fix a set $S \subseteq N \setminus H$. Let $\mu = F^{H,t}(S) = \mathbb{E}_{H' \sim H[t]} f(S \cup H') = \frac{1}{\binom{h}{t}} \sum_{H' \in H[t]} f(S \cup H')$. By definition, $H_1, \ldots, H_m$ are sampled from $H[t]$ without replacement. Since each $f(S \cup H')$ is bounded in $[0, f_{\max}]$, by Hoeffding's inequality for sampling without replacement (see, e.g., Proposition 1.2 in [1]), we have

$$\Pr\left[\left|\frac{1}{m} \sum_{i=1}^{m} f(S \cup H_i) - \mu\right| \geq \frac{\varepsilon}{2}\right] \leq 2 \exp\left(-\frac{m\varepsilon^2}{2f_{\max}^2}\right) \leq \delta' \tag{9}$$

given $m \geq \frac{2f_{\max}^2}{\varepsilon^2} \log \frac{2}{\delta'}$.

Assume that $|\frac{1}{m} \sum_{i=1}^{m} f(S \cup H_i) - \mu| \leq \frac{\varepsilon}{2}$ holds. We then consider the difference

$$\frac{1}{m} \sum_{i=1}^{m} \tilde{f}(S \cup H_i) - \frac{1}{m} \sum_{i=1}^{m} f(S \cup H_i) = \frac{1}{m} \sum_{H' \subseteq H} X_i$$

where

$$X_i = \tilde{f}(S \cup H_i) - f(S \cup H_i) = (\xi_{S \cup H_i} - 1) f(S \cup H_i)$$

is a random variable with mean $\mathbb{E}[X_i] = 0$ (by the unbiased property of noise) and is sub-exponential with parameters

$$(\nu f_{\max}, \quad \alpha f_{\max}).$$

Because the sets $H_1, \ldots, H_m$ are sampled without replacement, we have $H_i \neq H_j$ for $i \neq j$, so $S \cup H_i \neq S \cup H_j$. This means that the noise multipliers $\xi_{S \cup H_i}$ and $\xi_{S \cup H_j}$ are independent, so $X_i$ and $X_j$ are independent. Then, we apply Lemma B.1 to $\frac{1}{m} \sum_{i=1}^{m} X_i$ with $\alpha_* = \alpha f_{\max}$ and $\nu_* = \sqrt{m \nu^2 f_{\max}^2}$ to obtain

$$\Pr\left[\left|\frac{1}{m} \sum_{i=1}^{m} X_i\right| > \frac{\varepsilon}{2}\right] \leq 2 \exp\left(-\frac{m\varepsilon^2}{8\nu_*^2/m}\right) = 2 \exp\left(-\frac{m\varepsilon^2}{8\nu^2 f_{\max}^2}\right) \leq \delta' \tag{10}$$

given $m \geq \frac{8\nu^2 f_{\max}^2}{\varepsilon^2} \log \frac{2}{\delta'}$ and $0 \leq \frac{\varepsilon}{2} \leq \frac{\nu_*^2}{m\alpha_*} = \frac{\nu^2 f_{\max}^2}{\alpha f_{\max}} = \frac{\nu^2}{\alpha} f_{\max}$.

Taking a union bound over (9) and (10) and a union bound over all sets $S \subseteq N \setminus H$, we have with probability at least $1 - 2 \cdot 2^n \delta'$, for all sets $S \subseteq N \setminus H$, we have both $|\frac{1}{m} \sum_{i=1}^{m} f(S \cup H_i) - \mu| \leq \frac{\varepsilon}{2}$ and $|\frac{1}{m} \sum_{i=1}^{m} X_i| \leq \frac{\varepsilon}{2}$ hold, which implies

$$|\hat{F}^{H,t}(S) - F^{H,t}(S)| = \left|\frac{1}{m} \sum_{i=1}^{m} \tilde{f}(S \cup H_i) - \mu\right| \leq \varepsilon.$$

Let $\delta = 2 \cdot 2^n \delta'$. The $m$ has to satisfies

$$\begin{aligned}
m &\geq \max\left\{\frac{2f_{\max}^2}{\varepsilon^2} \log \frac{2}{\delta'}, \frac{8\nu^2 f_{\max}^2}{\varepsilon^2} \log \frac{2}{\delta'}\right\} \\
&= \max\{2, 8\nu^2\} \frac{f_{\max}^2}{\varepsilon^2} \log \frac{4 \cdot 2^n}{\delta} \\
&= \max\{2, 8\nu^2\} \frac{f_{\max}^2}{\varepsilon^2} \left(n \log 2 + \log \frac{4}{\delta}\right),
\end{aligned}$$

which is satisfied when $m \geq \max\{2, 8\nu^2\} \frac{f_{\max}^2}{\varepsilon^2} \left(n + \log \frac{4}{\delta}\right)$.

In order to sample $m$ sets $H_1, \ldots, H_m$ from $H[t]$ without replacement, the $h$ and $t$ have to satisfy

$$\binom{h}{t} \geq m.$$

By the inequality $\binom{h}{t} \geq \frac{n^t}{4t!}$ for $t \leq \sqrt{h}$ and letting $h = t^2$, we have

$$\binom{h}{t} \geq \frac{h^t}{4t!} \geq \frac{h^t}{4t^t} = \frac{t^t}{4} \geq \frac{2^t}{4} \geq m$$

when $t \geq \log_2(4m)$. $\qquad\square$

## B.2  Proof of Lemma 3.5 for Monotone Submodular Function

We follow the proof for the non-monotone case until (7), where we have

$$\mathbb{E}_H\Big[\max_{S\in\mathcal{I}_H} F^{H,t}(S)\Big] \;=\; \mathbb{E}_H\mathbb{E}_{H'\sim H[t]}\Big[f(O^* - \sigma(H) + H')\Big].$$

Because $f$ is monotone, we immediately have $f(O^* - \sigma(H) + H') \ge f(O^* - \sigma(H))$ and hence

$$\mathbb{E}_H\Big[\max_{S\in\mathcal{I}_H} F^{H,t}(S)\Big] \;\ge\; \mathbb{E}_H\Big[f(O^* - \sigma(H))\Big]$$

$$\text{by (8)} \;\ge\; f(O^*) - \frac{h}{r-h}f(O^*).$$

## C  Robustness of Measured Continuous Greedy: Proof of Lemma 4.1

---

**Algorithm 2:** Measured continuous greedy [9] with approximate value oracle

---

**Input :** Approximate value oracle $\hat{f}$ to a submodular function on ground set $N$. Matroid $\mathcal{I}$.

1 Let $n = |N|$, $\delta = n^{-4}$.

2 Initialize $t = 0$, $x(0) = \mathbf{0} \in [0,1]^n$.

3 **while** $t < 1$ **do**

4 $\quad$ Let $R(t)$ be the random set that contains each element $i \in N$ independently with probability $x_i(t)$.

5 $\quad$ For each $i \in N$, let $\hat{\omega}_i(t)$ be an estimate of the expected marginal value $\mathbb{E}[\hat{f}_{R(t)}(i)]$, obtained by taking the average of $\frac{10}{\delta^2}\log(2n)$ samples of $\hat{f}_{R(t)}(i)$.

6 $\quad$ Let $\hat{I}(t) = \arg\max_{I\in\mathcal{I}} \sum_{i\in I} \hat{\omega}_i(t)$ be a maximum-weight independent set.

7 $\quad$ Let $x(t+\delta)$ be the following: for every $i \in N$, $x_i(t+\delta) \leftarrow x_i(t) + \delta(1 - x_i(t))\hat{I}_i(t)$.

8 $\quad$ $t \leftarrow t + \delta$.

9 **end**

10 Use pipage rounding [4] (which does not require access to $\hat{f}$) to convert the fractional solution $x(1)$ to a discrete set $S \in \mathcal{I}$.

11 **return** $S$

---

In this section, we establish the robustness of the measured continuous greedy algorithm [9] (the full algorithm is given in Algorithm 2). Let $\hat{f}$ be an $\varepsilon$-approximate value oracle for $f$. The proof for the approximation ratios of the measured continuous greedy algorithm with exact value oracle is provided by Feldman [8]. We analyze how the $\varepsilon$-approximate value oracle $\hat{f}$ will affect the approximation ratios. The main step is Lemma C.4 which provides a replacement of Corollary 3.2.7 of [8]. Once this is established, we can apply the remaining arguments in Section 3.2.1 and Section 3.2.2 of [8] by replacing the discretization error of $O(n^3\delta^2)$ with the discretization and approximation error established in Lemma C.4.

Here, we use $F$ to denote the multilinear extension of the submodular $f$. In other words,

$$F(x) = \sum_{S\subseteq[n]} f(S) \prod_{i\in S} x_i \prod_{i\notin S}(1 - x_i), \quad \forall x \in [0,1]^n.$$

We use $\hat{F}$ to denote the multilinear extension with $f$ replaced with $\hat{f}$.

$$\hat{F}(x) = \sum_{S\subseteq[n]} \hat{f}(S) \prod_{i\in S} x_i \prod_{i\notin S}(1 - x_i), \quad \forall x \in [0,1]^n.$$

We require two well-known properties of the multilinear extension.

**Claim C.1.** *The partial derivatives of the multilinear extension $F$ satisfy:*

- $\partial_i F(x) = F(x \vee 1_i) - F(x \wedge 1_{\bar{i}})$.

- $\partial_i\partial_j F(x) = F(x \vee 1_i \vee 1_j) - F(x \vee 1_i \wedge 1_{\bar{j}}) - F(x \wedge 1_{\bar{i}} \vee 1_j) + F(x \wedge 1_{\bar{i}} \wedge 1_{\bar{j}})$.

At a high level, the measured continuous greedy algorithm works as follows. Let $\delta \in (0, 1)$ be defined such that $1/\delta$ is an integer. Let $\mathcal{P}$ be the matroid polytope. Given a point $x_i(t)$ at time $t$, we use samples to estimate $\partial_i \hat{F}(x(t))$ and then solve

$$y^*(t) \in \arg\max_{\mathcal{P}} \left\{ \sum_{i=1}^{n} \partial_i \hat{F}(x(t)) y_i \right\}.$$

Then we update $x_i(t + \delta) = x_i(t) + \delta(1 - x_i(t)) y_i^*(t)$.

Note that $x(1)$ is feasible since the update at time $t$ is bounded by $\delta y^*(t)$ which is feasible and thus $\delta(y^*(0) + y^*(1/\delta) + \ldots + y^*(1 - 1/\delta))$ is also feasible. Since the matroid polytope is downward-closed, we conclude that $x(1)$ is feasible.

We use $1_i$ to denote the vector whose $i$th coordinate is 1 and 0 otherwise and $1_{\bar{i}} = 1 - 1_i$. Let $\text{OPT} = \arg\max_{S \in \mathcal{I}} f(S)$ be an optimal solution. We let $\alpha = \frac{\varepsilon}{f(\text{OPT})}$.

**Lemma C.2.** *Suppose* $|f(S) - \hat{f}(S)| \leq \varepsilon = \alpha f(\text{OPT})$. *Then, for every $i$, we have* $|\partial_i F_i(x) - \partial_i \hat{F}_i(x)| \leq 2\alpha f(\text{OPT}) = 2\varepsilon$.

*Proof.* Directly follows from Claim C.1. $\qquad\square$

**Lemma C.3.** $\partial_i F(x) = \frac{F(x \vee 1_i) - F(x)}{1 - x_i}$.

*Proof.* This is a simple calculation. Indeed,

$$\partial_i F(x) = F(x \vee 1_i) - F(x \wedge 1_{\bar{i}}) = \frac{F(x \vee 1_i) - x_i F(x \vee 1_i) - (1 - x_i) F(x \wedge 1_{\bar{i}})}{1 - x_i}$$
$$= \frac{F(x \vee 1_i) - F(x)}{1 - x_i},$$

as desired. $\qquad\square$

We now establish the main lemma of this section.

**Lemma C.4.** *Suppose that* $|\hat{f}(S) - f(S)| \leq \alpha \cdot f(\text{OPT})$ *for all $S$. Then* $F(x(t + \delta)) - F(x(t)) \geq \delta \left( F(x(t) \vee 1_{\text{OPT}}) - F(x(t)) \right) - (4\delta\alpha n + n^3\delta^2) f(\text{OPT})$.

*Proof.* Let $z = x(t + \delta) - x(t)$ and consider the univariate function $g(s) = F(x(t) + sz)$. By a Taylor expansion, we have

$$F(x(t + \delta)) - F(x(t)) = g(1) - g(0) \geq g'(0) - \frac{1}{2} \max_{s \in [0,1]} |g''(s)|.$$

Taking derivatives, we have

$$g'(s) = \sum_{i=1}^{n} \partial_i F(x(t) + sz) z_i$$

and

$$g''(s) = \sum_{i=1}^{n} \sum_{j=1}^{n} \partial_i \partial_j F(x(t) + sz) z_i z_j.$$

We have the bound $|\partial_i \partial_j F| \leq 2n \cdot f(\text{OPT})$ (second item of Claim C.1) and $|z_i| \leq \delta$ so $|g''(s)| \leq 2n^3\delta^2 f(\text{OPT})$.

Now, we bound $g'(0)$. We have

$$
\begin{aligned}
g'(0) &= \sum_{i=1}^{n} \partial_i F(x(t)) z_i \\
&= \delta \sum_{i=1}^{n} \partial_i F(x(t))(1 - x_i(t)) y_i^*(t) \\
&\geq \delta \sum_{i=1}^{n} \partial_i \hat{F}_i(x(t)) \cdot (1 - x_i(t)) y_i^*(t) - 2\delta \alpha n \cdot f(\text{OPT}) \qquad \text{by Lemma C.2} \\
&\geq \delta \sum_{i \in \text{OPT}} \partial_i \hat{F}_i(x(t)) \cdot (1 - x_i(t)) - 2\delta \alpha n \cdot f(\text{OPT}) \\
&\geq \delta \sum_{i \in \text{OPT}} \partial_i F_i(x(t)) \cdot (1 - x_i(t)) - 4\delta \alpha n \cdot f(\text{OPT}) \qquad \text{by Lemma C.2} \\
&= \delta \sum_{i \in \text{OPT}} (F(x(t) \vee 1_i) - F(x(t))) - 4\delta \alpha n \cdot f(\text{OPT}).
\end{aligned}
$$

By submodularity, we have $\sum_{i=1}^{n}(F(x(t) \vee 1_i) - F(x(t))) \geq F(x(t) \vee 1_{\text{OPT}}) - F(x(t))$. To see the last inequality, let $\emptyset = S_0 \subset S_1 \subset \ldots \subset S_{|\text{OPT}|} = \text{OPT}$ be such that $|S_i \setminus S_{i-1}| = 1$. Then

$$
\begin{aligned}
F(x(t) \vee 1_{\text{OPT}}) - F(x(t)) &= \sum_{i=1}^{|\text{OPT}|} F(x(t) \vee 1_{S_i}) - F(x(t) \vee 1_{S_{i-1}}) \\
&= \sum_{i=1}^{|\text{OPT}|} F(x(t) \vee 1_{S_{i-1}+i}) - F(x(t) \vee 1_{S_{i-1}}).
\end{aligned}
$$

We can then iteratively apply Lemma C.5 to each summand. $\qquad \square$

**Lemma C.5.** *For any $i \neq j$, we have $F(x \vee 1_i \vee 1_j) - F(x \vee 1_i) \leq F(x \vee 1_j) - F(x)$.*

*Proof.* The inequality we want to prove can be written as

$$
\mathbb{E}_{S \sim x}[f(S + i + j) - f(S + i)] \leq E_{S \sim x}[f(S + j) - f(S)],
$$

which is true since $f$ is submodular. $\qquad \square$

## C.1 The Monotone Case

**Lemma C.6.** $F(x(1)) \geq \left[1 - 1/e - O(\alpha n + n^3 \delta)\right] \cdot f(\text{OPT})$.

*Proof.* From Lemma C.4, we have

$$
F(x(t + \delta)) \geq (1 - \delta) F(x(t)) + (\delta - 4\delta \alpha n - n^3 \delta^2) f(\text{OPT}).
$$

Let $C = (1 - 4\alpha n - n^3 \delta) f(\text{OPT})$ so that the above equation becomes

$$
F(x(t + \delta)) \geq (1 - \delta) F(x(t)) + \delta C.
$$

Unrolling the recursion, we have

$$
F(x(1)) \geq \sum_{i=0}^{1/\delta - 1} (1 - \delta)^i \delta C = \delta C \frac{1 - (1 - \delta)^{1/\delta}}{\delta} \geq C\left(1 - 1/e - \delta/2e\right),
$$

where we used Claim C.7 for the last inequality. Plugging in $C$ gives the claim. $\qquad \square$

**Claim C.7.** *If $x \leq 0.5$ then $(1 - x)^{1/x} \leq 1/e + x/2e$.*

*Proof.* First, by a Taylor expansion, we have $\log(1 - x) \leq -x + \frac{x^2}{2}$ which is valid for all $x \in (0, 1)$. We thus have $\frac{\log(1-x)}{x} \leq -1 + \frac{x}{2}$ so $(1 - x)^{1/x} \leq e^{-1} e^{x/2}$. Next, we use the numeric inequality $e^{x/2} \leq 1 + x$ which is valid for $x \leq 2$. So we conclude that $\frac{\log(1-x)}{x} \leq e^{-1}(1 + x/2)$. $\qquad \square$

From Lemma C.6, we obtain

$$
\begin{aligned}
F(x(1)) &\geq \left[1 - 1/e - O(\alpha n + n^3 \delta)\right] \cdot f(\text{OPT}) \\
&= \left[1 - 1/e - O(n^3 \delta)\right] \cdot f(\text{OPT}) - O(\varepsilon n) \\
&= \left[1 - 1/e - O(\tfrac{1}{n})\right] \cdot f(\text{OPT}) - O(\varepsilon n),
\end{aligned}
$$

with $\delta = n^{-4}$ in Algorithm 2, which proves Lemma 4.1 for the monotone case.

## C.2 The Non-Monotone Case

The following lemma can be established from following the proofs of Lemma 3.2.8, Lemma 3.2.9, Corollary 3.2.10, and Lemma 3.2.11 from [8] verbatim but replacing $O(n^3 \delta)$ in their argument with $O(\alpha n + n^3 \delta)$ (i.e. Lemma C.4).

**Lemma C.8.** $F(x(1)) \geq \left[1/e - O(\alpha n + n^3 \delta)\right] \cdot f(\text{OPT})$

This implies $F(x(1)) \geq \left[1/e - O(\tfrac{1}{n})\right] \cdot f(\text{OPT}) - O(\varepsilon n)$ and proves Lemma 4.1 for the non-monotone case.

# D Robustness of Double Greedy: Proof of Lemma 4.4

---

**Algorithm 3:** Double greedy [3] with approximate value oracle

---

**Input :** An approximate value oracle $\hat{f}$ for a non-negative submodular function $f : 2^N \to \mathbb{R}_+$.
1 Initialize $X_0 = \emptyset$, $Y_0 = N$.
2 Let $(u_1, \ldots, u_n)$ be an arbitrary order of the elements in $N$.
3 **for** $i = 1$ *to* $n$ **do**
4 $\quad$ Let $\hat{a}_i$ be the approximate value of $a_i = f_{X_{i-1}}(u_i) = f(X_{i-1} \cup \{u_i\}) - f(X_{i-1})$.
5 $\quad$ Let $\hat{b}_i$ be the approxiamte value of $b_i = -f_{Y_{i-1} \setminus \{u_i\}}(u_i) = f(Y_{i-1} \setminus \{u_i\}) - f(Y_{i-1})$.
6 $\quad$ Let $(\hat{p}_i, \hat{q}_i) = \begin{cases} (\frac{\hat{a}_i}{\hat{a}_i + \hat{b}_i}, \frac{\hat{b}_i}{\hat{a}_i + \hat{b}_i}) & \text{if } \hat{a}_i > 0 \text{ and } \hat{b}_i > 0 \\ (1, 0) & \text{if } \hat{a}_i > 0 \text{ and } \hat{b}_i \leq 0 \\ (0, 1) & \text{if } \hat{a}_i \leq 0. \end{cases}$
7 $\quad$ With probability $\hat{p}_i$, let $X_i = X_{i-1} \cup \{u_i\}$, $Y_i = Y_{i-1}$;
8 $\quad$ otherwise, let $X_i = X_{i-1}$, $Y_i = Y_{i-1} \setminus \{u_i\}$.
9 **end**
10 **return** $X_n$ (which equals $Y_n$).

---

This section proves the robustness of the double greedy algorithm [3] (given in Algorithm 3) against approximate value oracle.

**Lemma D.1.** *Suppose* $|\hat{a}_i - a_i| \leq \varepsilon$ *and* $|\hat{b}_i - b_i| \leq \varepsilon$. *Let* $\text{OPT} = \arg\max_{S \subseteq N} f(S)$. *The expected value of the solution* $X_n$ *returned by Algorithm 3 is at least*

$$
\mathbb{E}[f(X_n)] \geq \frac{1}{2} f(\text{OPT}) - \frac{3}{2} n\varepsilon. \tag{11}
$$

*Proof.* Define the following quantity:

$$
r_i = \max\{\hat{p}_i b_i, \hat{q}_i a_i\} - \tfrac{1}{2}(\hat{p}_i a_i + \hat{q}_i b_i). \tag{12}
$$

When the value oracle is exact, the probabilities $\hat{p}_i, \hat{q}_i$ are computed from the correct marginal values $a_i$ and $b_i$:

$$
(p_i, q_i) = \begin{cases} (\frac{a_i}{a_i + b_i}, \frac{b_i}{a_i + b_i}) & \text{if } \hat{a}_i > 0 \text{ and } b_i > 0 \\ (1, 0) & \text{if } a_i > 0 \text{ and } b_i \leq 0 \\ (0, 1) & \text{if } a_i \leq 0. \end{cases} \tag{13}
$$

In this case, it can be easily verified that $r_i = \max\{p_i b_i, q_i a_i\} - \tfrac{1}{2}(p_i a_i + q_i b_i) \leq 0$. Due to approximate oracle, $r_i = \max\{\hat{p}_i b_i, \hat{q}_i a_i\} - \tfrac{1}{2}(\hat{p}_i a_i + \hat{q}_i b_i)$ may be positive. [19] show that the loss of performance of the double greedy algorithm due to approximate oracle can be upper bounded by $\sum_{i=1}^n r_i$:

**Lemma D.2** (see Theorem 2.1 in [19] or Lemma 4.3 in [11]). *The expected value of the set $X_n$ returned by Algorithm 3 is at least:*

$$\mathbb{E}[f(X_n)] \geq \frac{1}{2}f(\mathrm{OPT}) - \mathbb{E}[\sum_{i=1}^{n} r_i]. \tag{14}$$

Then, it remains to upper bound $\mathbb{E}[\sum_{i=1}^{n} r_i]$, which we do in the following lemma:

**Lemma D.3.** *Suppose $|\hat{a}_i - a_i| \leq \varepsilon$ and $|\hat{b}_i - b_i| \leq \varepsilon$, then $r_i \leq \frac{3}{2}\varepsilon$.*

*Proof.* To simplify notations, we drop the subscript $i$, so $r = \max\{\hat{p}b, \hat{q}a\} - \frac{1}{2}(\hat{p}a + \hat{q}b)$. Consider three cases separately:

- $\hat{a} > 0$ and $\hat{b} > 0$. In this case, we have $\hat{p} = \frac{\hat{a}}{\hat{a}+\hat{b}}$, $\hat{q} = \frac{\hat{b}}{\hat{a}+\hat{b}}$, and

$$r = \frac{1}{\hat{a}+\hat{b}}\left(\max\{\hat{a}b, \hat{b}a\} - \frac{1}{2}(\hat{a}a + \hat{b}b)\right).$$

  Using the inequalities $a \leq \hat{a} + \varepsilon_a$, $b \leq \hat{b} + \varepsilon_b$, and $\hat{a} > 0, \hat{b} > 0$,

$$\begin{aligned}
r &\leq \frac{1}{\hat{a}+\hat{b}}\left(\max\{\hat{a}(\hat{b}+\varepsilon_b), \hat{b}(\hat{a}+\varepsilon_a)\} - \frac{1}{2}(\hat{a}(\hat{a}-\varepsilon_a) + \hat{b}(\hat{b}-\varepsilon_b))\right) \\
&= \frac{1}{\hat{a}+\hat{b}}\left(\hat{a}\hat{b} + \max\{\hat{a}\varepsilon_b, \hat{b}\varepsilon_a\} - \frac{1}{2}(\hat{a}^2 + \hat{b}^2 - \hat{a}\varepsilon_a - \hat{b}\varepsilon_b)\right) \\
&= \frac{1}{\hat{a}+\hat{b}}\left(\max\{\hat{a}\varepsilon_b, \hat{b}\varepsilon_a\} + \frac{1}{2}(\hat{a}\varepsilon_a + \hat{b}\varepsilon_b) - \frac{1}{2}(\hat{a}^2 + \hat{b}^2 - 2\hat{a}\hat{b})\right) \\
&\leq \frac{1}{\hat{a}+\hat{b}}\left(\max\{\hat{a}\varepsilon_b, \hat{b}\varepsilon_a\} + \frac{1}{2}(\hat{a}\varepsilon_a + \hat{b}\varepsilon_b)\right) \\
&= \max\{\hat{p}\varepsilon_b, \hat{q}\varepsilon_a\} + \frac{1}{2}(\hat{p}\varepsilon_a + \hat{q}\varepsilon_b) \\
&\leq \frac{3}{2}\varepsilon.
\end{aligned}$$

- $\hat{a} > 0$ and $\hat{b} \leq 0$. In this case, we have $\hat{p} = 1$, $\hat{q} = 0$, and

$$r = \max\{b, 0\} - \frac{1}{2}a$$

  On the one hand, $b \leq \hat{b} + \varepsilon_b \leq \varepsilon_b$. On the other hand, because $a + b \geq 0$ holds for any submodular function [3], we have $a \geq -b \geq -\varepsilon_b$. Therefore,

$$r \leq \max\{\varepsilon_b, 0\} - \frac{1}{2}(-\varepsilon_b) = \frac{3}{2}\varepsilon_b.$$

- $\hat{a} \leq 0$. In this case, we have $\hat{p} = 0$, $\hat{q} = 1$, and

$$r = \max\{0, a\} - \frac{1}{2}b$$

  On the one hand, $a \leq \hat{a} + \varepsilon_a \leq \varepsilon_a$. On the other hand, because $a + b \geq 0$ holds for any submodular function [3], we have $b \geq -a \geq -\varepsilon_a$. Therefore,

$$r \leq \max\{0, \varepsilon_a\} - \frac{1}{2}(-\varepsilon_a) = \frac{3}{2}\varepsilon_a.$$

All the three cases above give $r \leq \frac{3}{2}\max\{\varepsilon_a, \varepsilon_b\}$. □

Using Lemmas D.2 and D.3, we immediately obtain $\mathbb{E}[f(X_n)] \geq \frac{1}{2}f(\mathrm{OPT}) - \frac{3}{2}n\varepsilon$. □

# E Proof of Theorem 4.5

**Lemma E.1.** *Fix any set $S \subseteq N$. Sample a uniformly random set $H \subseteq N$ of size h. We have:*

$$\mathbb{E}_{H \sim N[h]}\big[F^{H,t}(S \setminus H)\big] \geq \mathbb{E}_{H \sim N[h]}\big[F^{H,t}(S)\big] - \frac{h}{|N| - h} \max_{S' \subseteq N: |S'| \leq |S| + h} f(S').$$

*Proof.* Let $H' \sim H[t]$ denote the random sampling of a subset $H'$ of $H$ with size $t$. By definition, $F^{H,t}(S) = \mathbb{E}_{H' \sim H[t]} f(S \cup H')$. So,

$$\begin{aligned}
\mathbb{E}_H[F^{H,t}(S) - F^{H,t}(S \setminus H)] &= \mathbb{E}_H \mathbb{E}_{H' \sim H[t]}\Big[f(S \cup H') - f((S \setminus H) \cup H')\Big] \\
&= \mathbb{E}_H \mathbb{E}_{H' \sim H[t]}\Big[f(S \cup H') - f((S \cup H') \setminus (H \setminus H'))\Big] \\
&= \mathbb{E}_{H'} \mathbb{E}_{H | H'}\Big[f(S \cup H') - f((S \cup H') \setminus (H \setminus H'))\Big],
\end{aligned}$$

where the notation $H | H'$ means sampling $H$ conditioning on $H'$. We note that, conditioning on $H'$, the distribution of the set $H \setminus H'$ is uniform across all subsets of $N \setminus H'$ of size $h - t$. So, letting $B = H \setminus H'$, we have

$$\begin{aligned}
\mathbb{E}_H[F^H(S) - F^H(S \setminus H)] &= \mathbb{E}_{H'} \mathbb{E}_{B \sim N \setminus H'}\Big[f(S \cup H') - f((S \cup H') \setminus B)\Big] \\
&\leq \mathbb{E}_{H'}\Big[\frac{|B|}{|N \setminus H'| - |B|} \max_{S' \subseteq S \cup H'} f(S')\Big] \qquad \text{by Lemma A.2} \\
&\leq \frac{h}{|N| - h} \max_{S' \subseteq N: |S'| \leq |S| + h} f(S').
\end{aligned}$$

$\square$

**Lemma E.2.** *For any set $S \subseteq N$. Sample a uniformly random set $H \subseteq N$ of size h, we have*

$$\mathbb{E}_{H \sim N[h]}\big[F^{H,t}(S)\big] \geq f(S) - \frac{h}{|N| - h} \max_{S': S \subseteq S' \subseteq N, |S'| \leq |S| + h} f(S').$$

*Proof.* Let $H' \sim H[t]$ denote a random subset of $H$ with size $t$. Then, we have

$$\begin{aligned}
\mathbb{E}_{H \sim N[h]}[F^{H,t}(S)] &= \mathbb{E}_{H \sim N[h]} \mathbb{E}_{H' \sim H[t]} f(S \cup H') \\
&= \mathbb{E}_{H' \sim N[t]}\big[f(S \cup H')\big] \\
&\geq f(S) - \frac{t}{|N| - t} \max_{S': S \subseteq S' \subseteq N, |S'| \leq |S| + t} f(S') \qquad \text{by Lemma A.3} \\
&\geq f(S) - \frac{h}{|N| - h} \max_{S': S \subseteq S' \subseteq N, |S'| \leq |S| + h} f(S').
\end{aligned}$$

$\square$

*Proof of Theorem 4.5.* The double greedy algorithm has approximation ratio $\alpha = 1/2$ and is $\beta(\varepsilon_1) = O(n\varepsilon_1)$-robust against $\varepsilon_1$-approximate value oracle by Lemma 4.4. Then, following the proof of Theorem 3.4, we have with probability at least $1 - \delta$ over the randomness of $\tilde{f}$, the expected value of the final solution ALG satisfies

$$\begin{aligned}
\mathbb{E}[f(\text{ALG}) \mid \mathcal{E}] &\geq \frac{1}{2} \mathbb{E}_{H \sim N[h]}\Big[\max_{S \subseteq N \setminus H} F^{H,t}(S)\Big] - O(n\varepsilon_1) \\
&\geq \frac{1}{2} \mathbb{E}_{H \sim N[h]}\big[F^{H,t}(O^* \setminus H)\big] - O(n\varepsilon_1) \qquad \text{(because } O^* \setminus H \subseteq N \setminus H) \\
&\geq \frac{1}{2}\Big(\mathbb{E}_{H \sim N[h]}\big[F^{H,t}(O^*)\big] - \frac{h}{n-h} f(O^*)\Big) - O(n\varepsilon_1) \qquad \text{by Lemma E.1} \\
&\geq \frac{1}{2}\Big(f(O^*) - 2\frac{h}{n-h} f(O^*)\Big) - O(n\varepsilon_1) \qquad \text{by Lemma E.2.}
\end{aligned}$$

Letting $\delta = \frac{1}{n}$ and $\varepsilon_1 = \frac{\varepsilon}{n} f(O^*)$, and taking into account the remaining $\delta$ probability, we have

$$\begin{aligned}
\mathbb{E}[f(\text{ALG})] &\geq (1-\delta)\mathbb{E}[f(\text{ALG}) \mid \mathcal{E}] + \delta \cdot 0 \\
&\geq \left(\frac{1}{2} - \frac{h}{n-h} - \frac{1}{n} - \varepsilon\right) f(O^*) \\
&= \left(\frac{1}{2} - O(\varepsilon)\right) f(O^*)
\end{aligned}$$

with $h = \Theta(\log^2(\frac{n}{\varepsilon}))$ and $n \geq \Omega(\frac{1}{\varepsilon}\log^2(\frac{n}{\varepsilon}))$. $\qquad\square$

# F   Discussion on High Probability Results

Our main results for noisy submodular maximization (Theorems 3.4, 4.2, 4.3, 4.5) are stated in terms of the expected value $\mathbb{E}[f(\text{ALG})] \geq (\alpha - o(1))f(O^*)$. We discuss high-probability results in this section.

A standard way to obtain high-probability results from expectation results is to repeat the randomized algorithm multiple times and output the best solution. In our case, this means repeating Algorithm 1 for $T$ times and picking the best set among the $T$ outputted sets $S_1, \ldots, S_T$. The challenge here is how to *compare two sets*, in order to pick the best one, using noisy values instead of the true values.

For **monotone** submodular functions, Huang et al. [16] provide a method to do noisy comparison. For any set $S \subseteq N$, define the following comparison surrogate function $f_0(S)$ and the noisy version $\tilde{f}_0(S)$:

$$f_0(S) = \frac{1}{|S|}\sum_{e \in S} f(S-e), \qquad \tilde{f}_0(S) = \frac{1}{|S|}\sum_{e \in S} \tilde{f}(S-e). \tag{15}$$

To compare two sets $S_1$ and $S_2$, we compare $\tilde{f}_0(S_1)$ and $\tilde{f}_0(S_2)$.

**Lemma F.1** ([16]). *Let $\varepsilon, \delta \in (0, 1/2)$. Suppose $|S| \geq \frac{\kappa}{\varepsilon}\log(\frac{2}{\delta})$ where $\kappa$ is the sub-exponential norm of the noise multiplier. Then,*

$$\Pr\left[\left|\tilde{f}_0(S) - f_0(S)\right| > \varepsilon f_0(S)\right] \leq \delta. \tag{16}$$

Now, suppose we repeat Algorithm 1 for $T$ times and obtain solutions $S_1, \ldots, S_T$. Let $X_i = \frac{f(S_i)}{f(O^*)} \in [0,1]$. By Theorem 4.2, the expected value of each solution satisfies $\mathbb{E}[f(S_i)] \geq (1-1/e-\varepsilon)f(O^*)$, namely $\mathbb{E}[X_i] \geq 1 - 1/e - \varepsilon$. According to Markov's inequality, $\Pr[X_i < 1 - 1/e - 2\varepsilon] = \Pr[1 - X_i > 1/e + 2\varepsilon] \leq \frac{\mathbb{E}[1-X_i]}{1/e+2\varepsilon} \leq \frac{1/e+\varepsilon}{1/e+2\varepsilon} \approx 1 - e\varepsilon$, hence

$$\Pr\left[\max_{i \in [T]} X_i < 1 - 1/e - 2\varepsilon\right] \leq (1 - e\varepsilon)^T \leq \delta$$

given $T \geq \frac{\log(1/\delta)}{e\varepsilon}$. Namely, with probability at least $1 - \delta$, we have

$$\max_{i \in [T]} f(S_i) \geq (1 - 1/e - 2\varepsilon)f(O^*). \tag{17}$$

Let $S_{\tilde{i}}$ be the best solution according to noisy comparison, namely $\tilde{i} = \arg\max_{i \in [T]} \tilde{f}_0(S_i)$. Let $i^* = \arg\max_i f(S_i)$. The true value of $S_{\tilde{i}}$ satisfies

$$\begin{aligned}
f(S_{\tilde{i}}) &\geq f_0(S_{\tilde{i}}) && \text{by monotonicity} \\
&\geq \frac{1}{1+\varepsilon}\tilde{f}_0(S_{\tilde{i}}) && \text{by Lemma F.1} \\
&\geq \frac{1}{1+\varepsilon}\tilde{f}_0(S_{i^*}) && \text{by the definition of } \tilde{i} \\
&\geq \frac{1-\varepsilon}{1+\varepsilon}f_0(S_{i^*}) && \text{by Lemma F.1} \\
&\geq \frac{1-\varepsilon}{1+\varepsilon}\left(1 - \frac{1}{|S_{i^*}|}\right)f(S_{i^*}) && \text{by Lemma A.1} \\
&\geq \frac{1-\varepsilon}{1+\varepsilon}\left(1 - \frac{1}{|S_{i^*}|}\right)\left(1 - \frac{1}{e} - 2\varepsilon\right)f(O^*) && \text{by (17).}
\end{aligned}$$

For maximizing a monotone submodular function, the size of each solution satisfies $|S_i| = r$. So, for the failure probability of $T$ noisy comparisons to be less than $\delta$, according to Lemma F.1 we need

$$r = |S_i| \geq \Omega\Big(\frac{\kappa}{\varepsilon} \log\big(\frac{2T}{\delta}\big)\Big) = \Omega\Big(\frac{\kappa}{\varepsilon} \log\big(\frac{2\log(1/\delta)}{\delta\varepsilon}\big)\Big).$$

We thus obtain the following high-probability result for noisy submodular maximization for monotone functions under matroid constraints:

**Corollary F.2.** *Fix $\varepsilon \in (0, \frac{2\nu^2}{\alpha})$. Suppose the matroid's rank $r \geq \Omega(\frac{1}{\varepsilon} \log^2(\frac{n}{\varepsilon}) + \frac{\kappa}{\varepsilon} \log\big(\frac{2\log(1/\delta)}{\delta\varepsilon}\big))$. By repeating the algorithm in Theorem 4.2 for $T = \frac{\log(1/\delta)}{e\varepsilon}$ times and outputting $S_{\bar{i}}$, we obtain an algorithm for maximizing monotone submodular functions under matroid constraints with noisy value oracle that, with probability at least $1 - \delta$, returns a solution satisfying*

$$f(S_{\bar{i}}) \geq \Big(1 - \frac{1}{e} - O(\varepsilon)\Big) f(O^*).$$

For **non-monotone** submodular functions, the above approach to obtaining high-probability result does not work because: (1) the $f(S_i) \geq f_0(S_i)$ step in the noisy comparison analysis no longer holds, and (2) the size of the solution $|S_i|$ can be less than the rank $r$ of the matroid, so the $(1 - \frac{1}{|S_i|})$ factor in the approximation ratio can be small. Another attempt on obtaining high-probability results for non-monotone functions could be to use some concentration analyses for submodular functions, such as [23]. However, the concentration analyses in prior works usually require the Lipschitz constant (the largest absolute marginal value) of the submodular function $f$ to be small compared to $f(O^*)$. It remains open how to obtain high-probability results for maximizing general non-monotone submodular functions under noise.

