# OpenReview forum: "A Unified Approach to Submodular Maximization Under Noise"
_NeurIPS.cc/2025/Conference — NeurIPS 2025 poster_

### Official Review · Reviewer_GM3f · 2025-06-19

**Clarity:** 3
**Significance:** 2
**Originality:** 3
**Rating:** 3
**Confidence:** 3

**Summary:**

This paper studies the problem of submodular maximization problems under a noisy oracle. In particular, it is assumed that the noise is random and unbiased, and that multiple queries of the submodular objective on the same subset is persistent. In that sense, the estimate of the same marginal gain/subset can't be made more accurate through averaging over multiple queries. To address this challenge, the authors construct a surrogate function to estimate marginal gains and propose algorithms for both monotone and non-monotone submodular maximization under this noise model.

**Questions:**

Please refer to my comments under the "Strengths And Weaknesses" section. If the authors are able to adequately address the concern regarding the correctness of the proof, I would be open to raising my score. More generally, I recommend that the authors carefully review the technical details to ensure the correctness of all proofs.

**Ethical Concerns:**

["NO or VERY MINOR ethics concerns only"]

**Final Justification:**

The authors have addressed my concerns regarding the correctness of the technical proofs, and I have updated my score accordingly.

**Limitations:**

yes

**Quality:**

1

**Strengths And Weaknesses:**

* Strength:
1. The paper addresses a challenging and practically relevant problem. Unlike prior work that assumes either deterministic persistent noise or unbiased noise with access to repeated queries, this work considers a more realistic and difficult setting: unbiased random noise with only one query per subset, where noise is persistent.

2. The proposed method is quite interesting. By estimating the marginal gain through the construction of a surrogate function, it might be possible to proposed unified algorithms for solving various submodular maximization problems.
3. The paper is well-written and easy to follow

* Weakness:
1. A major concern lies in the correctness of the technical proofs. While I did not verify every detail, I identified a potential issue in Lemma 3.3. Specifically, to apply the Hoeffding's inequality, the random variables $f(S\cup H_i)$ for different $i\in[m]$ should be independent. However, since $H_i$ are sampled without replacement, $f(S\cup H_i)$s are no longer independent. This mistake in the proof raised concerns about whether there exists other incorrect proofs in this paper.

---

> ### Author Rebuttal · Authors · 2025-07-30
>
> Thank you for noting the issue with Lemma 3.3.  As noted in the paper, one cannot use the usual Hoeffding inequality out of the box because the random variables are not independent.  However, Hoeffding also proved a version for sampling without replacement (see Proposition 1.2 of [1] below or Section 6 of Hoeffding’s paper [2]). Basically, the result says that the same bound holds even when sampling without replacement.  Using the notation in [1], we would just set $X_i = f(S \cup H_i)$, so $X_1, …, X_m$ are sampled from a finite population of $h$-choose-$t$ elements without replacement.  We will add this reference to the proof of Lemma 3.3.
>
> -----
>
> [1] Bardenet, Rémi, and Odalric-Ambrym Maillard. "Concentration inequalities for sampling without replacement." Bernoulli (2015): 1361-1385.
>
> [2] Hoeffding, Wassily. "Probability inequalities for sums of bounded random variables." Journal of the American statistical association 58.301 (1963): 13-30.

---

> > ### Author Response · Authors · 2025-08-06
> >
> > Dear Reviewer,
> >
> > Given that it is near the end of the discussion phase, we would like to ask if you have any other concerns with the paper. As we mentioned in our rebuttal, the issue you mentioned can be easily remedied since there is a version of Hoeffding's Inequality that deals with sampling without replacement.
> >
> > Thanks again for taking a look at our paper!

---

> > ### Comment · Reviewer_GM3f · 2025-08-07
> >
> > Thank you very much for your response. I have reviewed the references you provided and found that a version of Hoeffding's inequality does indeed exist for the case of sampling without replacement. However, the version presented in the paper does not appear to be correct. I have raised my score accordingly, but I still encourage the authors to carefully verify the technical details to further improve the rigor and quality of the paper.

---

> > > ### Author Response · Authors · 2025-08-07
> > >
> > > Thank you for raising the score!  We want to confirm that our use of Hoeffding’s inequality for sampling without replacement is correct.
> > >
> > > - Suppose we have a finite population of $M$ elements $x_1, \ldots, x_M$, with population mean $\mu = \frac{1}{M} \sum_{i=1}^M x_i$.  These elements are lower bounded by $a$ and upper bounded by $b$.
> > >
> > > - We sample $m$ elements without replacement from this set. Let the samples be $X_1, \ldots, X_m$.
> > >
> > > - Proposition 1.2 of [1] (Bardenet & Mailard, 2015) says that the sampled mean satisfies $\Pr[ \frac{1}{m} \sum_{i=1}^m X_i - \mu \ge \epsilon ] \le \exp\Big( - \frac{2m\epsilon^2}{(b-a)^2} \Big)$.
> > >
> > > In our proof of Lemma 3.3, we have a finite population of $M = \binom{h}{t}$ elements, where each element (indexed by $H’$) is $x_{H’} = f(S\cup H’)$. By “sampling $m$ subsets $H_1, …, H_m$ without replacement” we mean sampling $m$ subsets $H_1, …, H_m$ such that they are distinct (they can still overlap with each other).  Let $X_i = f(S \cup H_i)$.  This corresponds to sampling $m$ elements $X_1, …, X_m$ from the population of $\binom{h}{t}$ elements without replacement.  We have $a = 0$ and $b = f_{\max}$.  We use the Hoeffding’s inequality with $\epsilon/2$ instead of $\epsilon$, and use it twice, which directly gives us Equation (9).

---

### Official Review · Reviewer_u1Du · 2025-06-22

**Clarity:** 3
**Significance:** 3
**Originality:** 3
**Rating:** 5
**Confidence:** 4

**Summary:**

A fundamental problem in optimization is maximizing a submodular set function. When having access to a value oracle (i.e. an oracle giving the value of the set function for any set *precisely*), approximation algorithms for the maximization problem subject to cardinality/matroid constraints are known. The paper under consideration studies this problem with a slight twist: the value oracle is noisy, i.e. we observe a noisy version $\tilde{f}(S)$ instead of $f(S)$, and we still would like to find a set which (approximately) maximizes the function's value.
The noise under consideration is multiplicative, i.e. for any set $S$, we have $\tilde{f} (S) = \epsilon_S \cdot f(S)$ for noise $\epsilon_S$.

The paper derives a black-box reduction from the noisy to the exact setting. More precisely, it states a meta-algorithm which satisfies that "under some assumptions", any approximation algorithm for the exact setting can be used to obtain the same approximation guarantee for the noisy case.
From a technical point of view, the meta-algorithm is based on a randomized surrogate function which allows to estimate the value of the function $f$ with a reasonable error.

Using their reduction, the authors then derive (tight) approximation guarantees for the noisy setting subject to cardinality and matroid constraints in the monotone and non-monotone regime by applying their meta-algorithm to several known approximation algorithms.

**Questions:**

* Based on one of the weaknesses mentioned above: What is the intuition behind the sub-exponential noise? What are the limitations of the approach towards general noise?

* The robustness assumption in Definition 3.1. seems restrictive at first glance. Is this really the case (even though several approx. algorithms fulfill this definition)? Could this be extended to less restrictive definitions as well?

* Is your approach also usable for, say, knapsack constraints or similar?

**Ethical Concerns:**

["NO or VERY MINOR ethics concerns only"]

**Final Justification:**

As said, I keep my positive score.

**Limitations:**

No negative societal impact expected.

**Paper Formatting Concerns:**

Paper is well-written and formatted.

**Quality:**

3

**Strengths And Weaknesses:**

*Strengths*:
The general framework is powerful in a sense to deliver tight approximation guarantees in the settings under consideration. It extends previous work (where surrogate functions to approximate the value of the submodular set functions where mainly deterministic) in a suitable way by randomization.
This allows the authors to provide a unified framework for several problem specifications (matroids/monotone/...) which were previously handled by more tailored approaches.
That said, the meta-algorithm is very clean -- in my opinion an additional plus of the construction in the paper.

*Weaknesses*:
From a purely technical perspective, the contribution of the paper seems to work in the direction of the natural extension of previous work. Working with surrogate functions was already known before and also the choice of the randomized surrogate function in the paper seems to be not super unexpected. In addition, the distribution of the noise is limited to sub-exponential distributions which ensure desirable concentration properties. This further weakens the results of the paper as it remains open if the approach can be extended to general noise as well.

---

> ### Author Rebuttal · Authors · 2025-07-30
>
> Thanks for your review and positive assessment!
>
> > (Weakness 1) Randomized surrogate function seems to be not super unexpected.
>
> We agree that the idea of using randomization is not super surprising in and of itself. However, we still find it interesting that a simple randomized framework is already sufficient to recover and improve existing results while also allowing us to easily prove new results. It also allows us to reduce the problem of designing and analyzing algorithms in the noisy setting to checking robustness of existing approximation algorithms. It was not clear from prior work that this decoupling was possible.
>
> > (Question 1) What is the intuition behind the sub-exponential noise? What are the limitations of the approach towards general noise?
>
> We would need some assumption on the distribution in order to have some result. Without any assumptions, we could have very bad distributions. For example, consider a case where there are n items and we just want to pick one. One item is very valuable but extremely noisy and will return 1 most of the time. All other items have value 1. It would not be possible for any algorithm to distinguish between these items. In addition, we also note that our sub-exponential assumption is already more general than prior work such as [11, 14].
>
> > (Question 2) The robustness assumption in Definition 3.1. seems restrictive at first glance. Is this really the case (even though several approx. algorithms fulfill this definition)? Could this be extended to less restrictive definitions as well?
>
> We suspect that the robustness definition is not an extremely strong assumption but we don’t yet have a formal argument for it. Intuitively, if you ran an approximation algorithm but the values were off by $\epsilon$ then, by virtue of it being an approximation algorithm, you would still expect a good approximation. It is an interesting question to see if one can formalize a connection between approximation algorithms and robustness.
>
>
> > (Question 3) Is your approach also usable for, say, knapsack constraints or similar?
>
> Great question! This is a great open question for future work. The analysis of our meta algorithm does rely on the fact the constraint is a matroid constraint.  Our analysis uses a basis-exchange property for matroids, which does not directly extend to more general constraints (e.g., knapsack constraint).

---

> > ### Comment · Reviewer_u1Du · 2025-08-04
> >
> > I thank the authors for their response and clarification. That said, I keep my positive impression of the paper and its contributions.

---

### Official Review · Reviewer_ZGB3 · 2025-06-27

**Clarity:** 3
**Significance:** 3
**Originality:** 3
**Rating:** 5
**Confidence:** 3

**Summary:**

Submodular maximization under noise is the problem of maximizing submodular function $f \colon 2^N \to \mathbb{R}_{\ge 0}$ with access to only a noisy oracle that returns $\tilde{f}(S) = \xi_S f(S)$ for each $S \subseteq N$, where $\xi_S$ is a stochastic and persistent sub-exponential noise with mean $1$. This paper proposes a unified framework for obtaining an algorithm with the same approximation ratio guarantee from any "robust" existing algorithm. Using this framework, the present paper proposes a $(1-1/e)$-approximation algorithm for the monotone and matroid case, $1/e$-approximation for the non-monotone and matroid case, and $1/2$-approximation for the non-monotone and unconstrained case.

**Questions:**

No question.

**Ethical Concerns:**

["NO or VERY MINOR ethics concerns only"]

**Final Justification:**

I did not have any concern, and I keep my positive score.

**Limitations:**

Yes

**Quality:**

3

**Strengths And Weaknesses:**

Strengths:
- The theoretical results are very good. It resolves the open question of whether it is possible to obtain the same approximation ratio even under the noisy setting under several constraints.
- The idea behind the proposed algorithm is interesting. As existing studies on this topic, the proposed algorithm maximizes a surrogate function instead of maximizing the noisy objective function itself. The main idea of this paper is to estimate the function value $f(S)$ by sampling values $f(S \cup H')$ for random $H' \subseteq H$ with $|H'| = t$. By using the smoothing lemma (Lemma 3.5), maximizing the surrogate function leads to a good approximation to maximizing the original function. This idea is very clean and can be applied to many other settings of submodular maximization. I have not read all the proofs in the appendices and do not understand why this simple approach overcomes the barrier that the existing papers had, but the proofs of Lemma 3.5 and Appendix A seem to use submodularity in a good way.
- The paper is clearly written.

---
Weaknesses:
- The definitions of submodularity in the beginning part of this paper are wrong. In the Introduction section, "$i \in T \setminus S$" should be replaced by "$i \in N \setminus T$". In the beginning of Section 2, "any element in $x \in N$" should be replaced by "any element in $x \in N \setminus B$. I believe that these are just typos, but if these wrong definitions are used anywhere in the proof, it would be problematic.

---

> ### Author Rebuttal · Authors · 2025-07-30
>
> Thanks for reviewing the paper and spotting the typo! We will fix it.

---

### Official Review · Reviewer_WQvx · 2025-06-30

**Clarity:** 2
**Significance:** 2
**Originality:** 2
**Rating:** 4
**Confidence:** 1

**Summary:**

This paper studies submodular function maximization under a persistent noisy value oracle model, where repeated queries to the same set yield a fixed noisy value. Building upon prior work [11, 14], the authors propose a meta-algorithm that systematically converts any robust noiseless submodular maximization algorithm into one that works under persistent noise, while preserving approximation guarantees. By instantiating this framework with measured continuous greedy and double greedy algorithms, the paper recovers or improves state-of-the-art approximation ratios across cardinality, matroid, and unconstrained settings for both monotone and non-monotone functions.

**Questions:**

See the questions in weakness section.

**Ethical Concerns:**

["NO or VERY MINOR ethics concerns only"]

**Final Justification:**

I am giving the paper a positive score; however, my assessment is largely an educated guess.

**Limitations:**

Yes.

**Paper Formatting Concerns:**

There is no formatting concerns

**Quality:**

3

**Strengths And Weaknesses:**

Strengths:
1. Introduce a meta-algorithm that robustly extends noiseless submodular maximization algorithms to the persistent noise setting, achieving near-optimal guarantees: a tight (1 -\frac{1}{e})-approximation for monotone submodular maximization under matroid constraints, a 1/2-approximation for unconstrained non-monotone maximization (tight even under noise), and a 1/e-approximation for non-monotone maximization under matroid constraints. This includes the first theoretical guarantees for noisy non-monotone submodular maximization.
2. Employ a novel randomized surrogate function technique that overcomes the limitations of prior deterministic smoothing approaches, allowing a unified treatment of various constraints while inheriting strong guarantees from continuous greedy and double greedy algorithms.
3. Significantly broaden the applicability of noise-aware submodular maximization by generalizing previous algorithms that were limited to monotone functions and specific constraints, providing tight approximation guarantees under realistic persistent noise models.

Weakness:
1. Although the paper is relatively complete in theoretical analysis, the core method is mainly based on the robustness analysis of existing continuous greedy and double greedy algorithms, lacking essential breakthroughs in existing work. In particular, the surrogate function design and sampling mechanism proposed in the framework have obvious inheritance relationships with previous works such as [11][14]. It is recommended that the author clearly explain the substantial innovation compared with these works.
2. The results shown in Table 1 of the paper have improved previous work in some scenarios. In the non-monotonic case, although it fills the theoretical gap of unconstrained and pseudo-matrix constraints, it is difficult to judge its practical value without comparing it with the empirical performance of heuristic methods.
3. Theorem 4.2 requires that the rank constraint satisfies $r \geq \Omega(\varepsilon^{-1} \log^2 n)$. Taking the typical parameters n=10^4 and ε=0.1 as an example, this means that $ r\geq 500$ is required to ensure that the theoretical result holds. However, many practical applications usually involve low-rank constraints (such as r=10). Does the theoretical guarantee of the framework completely fail?
4. Although it is a theoretical paper, since it claims to propose a "widely reusable unified algorithm framework", its actual effects should be demonstrated through small-scale experiments or simulations to demonstrate its "practical applicability".

---

> ### Author Rebuttal · Authors · 2025-07-30
>
> Thanks for your review.  We would like to address the weaknesses you pointed out.
>
> __(Weakness 1)__
> As you mentioned, one of the core methods is based on the robustness analysis of existing algorithms.  However, we think this is actually one of the main advantages of our work.  Previous work [11, 14] defined surrogate functions and implicitly used robustness arguments.  However, their analysis mixes these two components and is very specific to their setting.  It is unclear whether or not these two components can be separated to obtain a more general reduction.  In contrast, we construct a new surrogate function that enables these two parts to be analyzed separately, establishing a clean reduction from the noisy submodular maximization problem to the robustness analysis of existing algorithms.  Such a new surrogate function requires randomization that is not present in previous work.
>
> __(Weakness 2)__
> As with prior work on this topic (specifically [11, 14]), and many prior works on submodular optimization published in NeurIPS, our goal is to advance the theoretical understanding of submodular optimization with noise. With that said, we also agree with the reviewer that the empirical comparison with heuristic methods may be helpful to demonstrate the practical value of our algorithm.  So, we have implemented our algorithm and compared it with heuristic methods; please see the end of our rebuttal for details.
>
> __(Weakness 3)__
> Yes, the rank constraint is a current limitation of our work.  When the rank $r$ is very small, it is unclear whether the surrogate function approach (using auxiliary subsets to denoise the function) still works.  Nonetheless, we note that our work already improves upon the dependency of the rank in prior work such as [14].  In [14], they have two regimes.  The regime which is the most comparable is when $n \geq 1/\epsilon^4$ (Theorem 5.2 of [14]).  For reference, note that we require $n \geq 1/\epsilon$ so our requirement on $n$ is better (their other regime actually requires $n$ to be exponential in $poly(1/\epsilon)$). In this case, their Theorem 5.2 requires $r \geq n^{1/3} \geq 1 / \epsilon^{4/3}$.  So our dependency on the rank $r$ is exponentially better than [14].
>
> For completeness, we will note that in [14], if they assume $n$ is exponential in $poly(1/\epsilon)$ (although it is unclear what the degree of the polynomial is), then $r \geq 1/\epsilon$ is sufficient. In this regime, their dependency on the rank is better but we reiterate that this requires that $n$ is very large.
>
> __(Weakness 4)__
> As discussed earlier, our main focus is on a better theoretical understanding of submodular optimization with noise. To complement the theoretical perspective, we provide experiments to demonstrate the “practical applicability” of our algorithms.  Please see the details below.
>
>
> __Added experiments:__
> To address Reviewer WQvx’s concerns in Weaknesses 2 and 4, we provide some experimental results.  We focus on the simple yet underexplored case of unconstrained non-monotone submodular maximization.  We consider an example where the submodular function has additive weights and quadratic cost in size: $f(S) = \sum_{i\in S} w_i - c |S|^2$, with $w_i \sim \mathrm{Uniform}[0, 20]$ and $c = 0.1$, with groundset size $n=100$. To match the setting in our paper, we ensured $w_i$ were chosen so that $f$ was non-negative (although this does not impact the results). The noisy value is $\tilde f(S) = \xi_S f(S)$ where $\xi_S \sim \mathrm{Normal}(\mu=1, \sigma^2=0.1)$.  We compare four algorithms against the optimal value benchmark:
>
> (1) Double greedy (DG) algorithm with exact value oracle (Buchbinder et al, 2015).  This is a worst-case optimal polynomial-time algorithm in the noiseless setting.
>
> (2) Random subset: pick a random subset of size $n/2$.  This trivial algorithm is only for reference.
>
> (3) Double greedy (DG) algorithm that uses the noisy value oracle directly. This is a natural heuristic algorithm.  Reviewer WQvx suggested we compare against heuristic algorithms.
>
> (4) Our algorithm in Theorem 4.5, which is the surrogate-function-based meta-algorithm instantiated with double greedy.   After simple tuning, we set the parameters to h = 20, t = 4, and vary m.
>
> Results are shown in the following table. We fix a function $f$, run the randomized algorithms for 100 simulations, and show the mean values (E[ALG] / OPT) and standard deviations.
>
> |    Optimal value    |  DG with exact oracle  |    Random subset    |  DG with noisy oracle  |  Our algorithm (m=50)  |  Our algorithm (m=200) |
> | :-----------------: | :--------------------: | :-----------------: | :--------------------: | :--------------------: | :--------------------: |
> |   normalized to 1   |    0.940 (+/- 0.017)   |  0.527 (+/- 0.054)  |    0.557 (+/- 0.048)   |    0.659 (+/- 0.047)   |    0.742 (+/- 0.045)   |
>
> We see that the heuristic algorithm (DG with noisy oracle) does not perform well in this example; its performance is similar to a random subset. Our algorithm significantly outperforms the heuristic algorithm, with 18% improvement with m = 50 and 33% with m = 200.

---

> > ### Comment · Reviewer_WQvx · 2025-08-07
> >
> > Thank you for the reply and for providing the experiments. However, following up on the response to W2, I would appreciate it if the authors could provide more details explaining where the novelty lies in extending the existing continuous greedy and double greedy algorithms to their problem.
> >
> > Certainly, there are many breakthrough papers that employ continuous greedy or double greedy algorithms in novel ways. My point is that the authors should clarify and emphasize how their extension is novel and why it is worthy of publication in this venue.

---

> ### Author Response · Authors · 2025-08-07
>
> Thank you for the question.
>
> To reiterate, we view the main contribution and novelty of our work as providing a reduction from the noisy setting to the non-noisy setting. This reduction allows us to reuse existing algorithms completely out of the box.  We are not just “extending the existing continuous greedy and double greedy algorithms to our problems” – any existing algorithm that is robust in the non-noisy setting can be plugged into our framework. For example, for the matroid-constrained monotone case, one can also use the local search algorithm (instead of continuous greedy) in our meta-algorithm.  The known (1-1/e) approximation guarantee of local search in the non-noisy setting directly extends to the noisy setting.  In contrast, the algorithms and analyses in [11][14] are specific to their settings: for example, [14] uses the local search algorithm with some special deterministic surrogate function and derives a sub-optimal (1-1/e)/2 approximation ratio for the matroid-constrained monotone case via specific analysis.  Our work directly improves upon [14], as well as deriving new results, via a unified analysis.
>
> Making this reduction work is non-trivial (otherwise, [11, 14] would be trivial). As we mentioned in our initial response, we needed a new surrogate function to overcome the limitations of [11, 14]. It turns out that the idea is simple: use a random surrogate function. While this is simple, it was not considered in prior work and turns out to be extremely useful.
>
> In addition, how to define the random surrogate function requires some care. The most naive way to do this would be to sample a small (but large enough) set $H$ and then obtain a surrogate function by considering $E[f(S \cup H’)]$ where $H’$ is a uniformly random subset of $H$. We do not know if this works or not as our proof would not work; this breaks when trying to handle the matroid constraints. The crucial idea here is to use fixed-size subsets of $H$ and combine this with a generalized version of the basis exchange property ([5] in our paper). The connection to [5] may also be of interest as we are not aware of any works in submodular optimization that make use of this version of the basis exchange property.

---

> > ### Comment · Reviewer_WQvx · 2025-08-07
> >
> > In the paper, I can not find runtime analysis. Would you provide runtime (and/or query number) of the algorithm in comparison with related works?

---

> > > ### Author Response · Authors · 2025-08-07
> > >
> > > We mentioned our runtime in Theorem 3.4: it is $m \cdot Q$ where $Q$ is the query complexity of the non-noisy algorithm and $m$ is a parameter that we mentioned in Lemma 3.3. In Lemma 3.3, if we fix the error parameter $\epsilon$ (say, we want to achieve a $1-1/e-\epsilon$ approximation ratio) to be a constant, then $m \approx O(n / \epsilon^2) = O(n)$.  This means that we have an $O(n)$ blow-up: the query complexity for the noisy setting is $O(n)$ times the query complexity for the non-noisy setting. We will clarify that $m \approx O(n)$ after Theorem 3.4.
> > >
> > > The comparison between our work and [11, 14] does not have a conclusive answer because it depends on the exact settings. If we focus on the simplest problem of maximizing a monotone submodular function subject to a cardinality constraint, then the query complexities are:
> > >
> > > - For [11], their analysis gives a running time of $n^{O(1)}$ where the $O(1)$ seems to be at least 25 or so. They did not explicitly specify the running time in any of their results. Rather, we inferred this from their proof, where they mention they require a smoothing set of size at least $25 \log n$. It is likely that their $O(1)$ constant can be optimized using a sampling approach similar to what we did, but [11] did not do this in their paper.
> > >
> > > - For [14], they mentioned that the running time is $r^2 n^3$ when the cardinality $r$ is large (their Theorem 4.6), and $r^2 n^{3/2}$ when the cardinality $r$ is small (their Theorem 4.1).  [14] uses the local search algorithm, which has $r^2 n$ query complexity in the non-noisy setting.  So, [14] has a $n^2$ blow-up for the large cardinality case.
> > >
> > > - Our algorithm runs in time $O(r n^2)$ where $r$ denotes the cardinality constraint. It comes from the $Q = O(rn)$ query complexity of the non-noisy greed algorithm times the $m = O(n)$ blow-up. So clearly, this improves upon the previous algorithms [11, 14] for the large cardinality case.
> > >
> > > In particular, note that our algorithm always incurs a $O(n)$ blow-up in the query complexity from the non-noisy setting to the noisy setting. However, in the worst case, prior work requires at least $n^2$ blow-up.  We view this as another advantage of our unified framework.

---

> ### Comment · Reviewer_WQvx · 2025-08-09
>
> Thanks for the response. I now have a clearer understanding of the paper’s contribution. I will adjust the score accordingly.

---

### Note · Authors · 2025-08-14

We would like to thank all the reviewers again for their time in reviewing our paper. We wanted to take this opportunity to summarize the main outcome of the discussion which seemed to have resolved the reviewers’ concerns, as there were no more questions after our responses to them.

Reviewer WQvx had asked several questions about our paper (e.g. practicality, novelty, contribution, time complexity). The reviewer seemed satisfied with our responses (there were no more questions) and had mentioned that they would adjust their score in their final response to us. (It seems to us the score has not changed but perhaps we are not able to see changes in the score.)

Reviewer GM3f pointed out a minor issue in our proofs regarding the application of Hoeffding’s Inequality. As we pointed out in our response, this can be easily fixed by using the “sampling without replacement” version of Hoeffding’s Inequality which already appears in Hoeffding’s original paper. In our paper, the only adjustment we need to make is to mention this in our statement of Hoeffding’s Inequality. We have also added a more detailed explanation in a follow up response that the application is fine. In their initial review, the reviewer also mentioned that they would adjust their score if this issue could be fixed.

Thanks again!

---

### Decision · Program_Chairs · 2025-09-17

**Decision:**

Accept (poster)

**Comment:**

The paper studies submodular maximization in the setting where we are given only noisy evaluations and the noise is persistent in the sense that multiple calls to the evaluation oracle with the same set return the same value. This model was introduced and studied in prior work, which also designed an algorithm with an optimal 1-1/e approximation for monotone submodular maximization with a cardinality constraint and a sub-optimal (1-1/e)/2 approximation for a matroid constraint. The main contribution of this work is an algorithm that provides a black-box way of turning an algorithm for the setting without noise with certain properties into an algorithm for the noisy setting with only an o(1) loss in the approximation. By combining this algorithm with the continuous greedy algorithm and the double greedy algorithm, the paper obtains improved approximation guarantees for a matroid constraint and the unconstrained setting.

The reviewers appreciated the theoretical contributions and found the main results to be strong. The reviewers also raised several concerns, most of which were sufficiently addressed by the author response and the subsequent discussion. One of the remaining concerns was that the novelty of the techniques is somewhat limited. Following the discussion, there was sufficient support among the reviewers for accepting the paper, and the reviewer with a more negative evaluation noted that the discussion addressed their concerns.